# MIMICKING THE PHYSICIST'S EYE : A VLM-CENTRIC APPROACH FOR PHYSICS FORMULA DISCOVERY

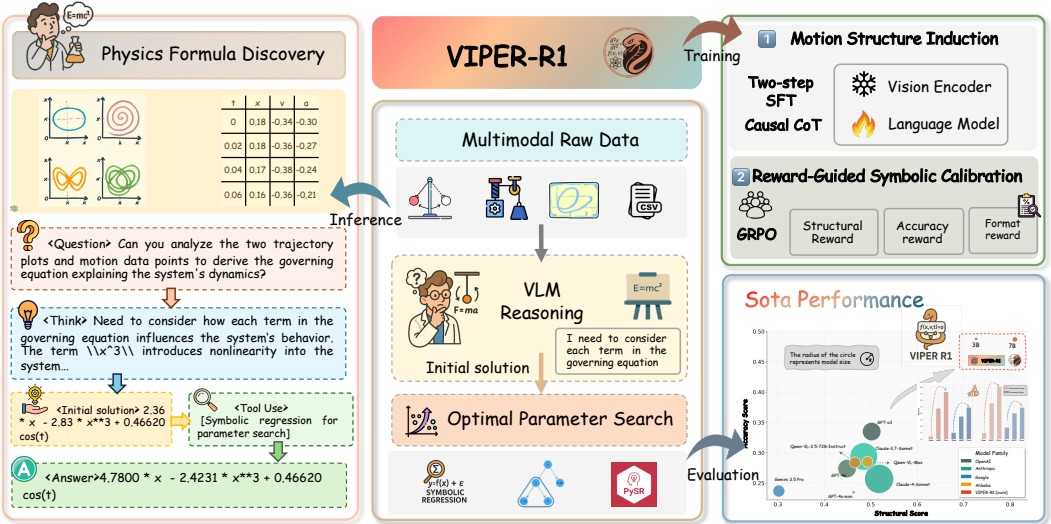

Figure 1: Overview of **VIPER-R1**, a multimodal framework for physics formula discovery. The model is trained via Motion Structure Induction (MSI) with Causal CoT supervision and Reward-Guided Symbolic Calibration (RGSC) for structural refinement. During inference, VIPER-R1 acts agentically by invoking an external symbolic regression tool for Symbolic Residual Realignment (SR²), reconciling symbolic hypotheses with empirical data. The model achieves state-of-the-art performance in both structural and accuracy scores on the PhysSymbol dataset.

## ABSTRACT

Automated discovery of physical laws from observational data in the real world is a grand challenge in AI. Current methods, relying on symbolic regression or Large Language Models (LLMs), are limited to uni-modal data and overlook the rich, visual phenomenological representations of motion that are indispensable to physicists. This "sensory deprivation" severely weakens their ability to interpret the inherent spatio-temporal patterns within dynamic phenomena. To address this gap, we propose **VIPER-R1**, a multimodal model that performs **V**isual **I**nduction for **P**hysics-based **E**quation **R**easoning to discover fundamental symbolic formulas. It methodically integrates visual perception, trajectory data, and symbolic reasoning to simulate the scientific discovery process. The model is trained via a curriculum of Motion Structure Induction (MSI), using supervised fine-tuning to interpret kinematic phase portraits and construct hypotheses guided by a Causal Chain of Thought (C-CoT), followed by Reward-Guided Symbolic Calibration (RGSC) to purify the formula's structure with reinforcement learning. During inference, the trained **VIPER-R1** acts as an agent: it first posits a high-confidence symbolic ansatz, then proactively invokes an external symbolic regression tool to perform Symbolic Residual Realignment (SR²). This final step, analogous to a physicist's perturbation analysis, reconciles the theoretical model with empirical data. To support this research, we introduce PhysSymbol, a new 10,000-instance multimodal corpus. Experiments show that **VIPER-R1** consistently outperforms state-of-the-art Vision Language Models (VLMs) baselines in accuracy and interpretability, enabling more precise discovery of physical laws.

# 1 INTRODUCTION

The automated discovery of fundamental physical laws in the form of equations from observational data stands as a grand challenge at the intersection of artificial intelligence and the natural sciences (Udrescu & Tegmark, 2020; Wang et al., 2023a; Hu et al., 2025). This endeavor is pivotal for augmenting human scientific intuition and accelerating the pace of discovery by uncovering novel principles within vast, high-dimensional datasets (Lu et al., 2024; Reddy & Shojaee, 2025). Recent advances have established two parallel yet distinct research tracks: sophisticated symbolic regression (SR) algorithms that navigate immense combinatorial spaces to identify fitting equations (La Cava et al., 2021; Cranmer, 2023), and the emergence of Large Language Models (LLMs) demonstrating remarkable ability to perform in-context symbolic reasoning from textual data (Ma et al., 2024a; Grayeli et al., 2024; Shojaee et al., 2025a). While both approaches have laid critical foundations, they share a disconnect with the actual process of human scientific inquiry, operating without a key perceptual faculty that is central to human discovery.

This limitation can be seen as a form of "sensory deprivation," where reliance on uni-modal symbolic data blinds models to the rich visual representations that physicists routinely exploit. Human scientific reasoning is inherently multimodal: physicists interpret visual patterns in phase portraits to infer conservation laws, recognize decay envelopes to hypothesize damping forces, and identify superposition effects to constrain theoretical possibilities (Strogatz, 2001). Such visual intuition provides powerful pre-symbolic heuristics for navigating the vast space of candidate theories.

Recent advances in LLM-based scientific discovery partly address these issues. LLM-SR (Shojaee et al., 2025a) generates equation hypotheses from embedded scientific knowledge, while frameworks like Scientific Generative Agents (Ma et al., 2024a) pair LLM-based generation with simulation validation. Yet these methods still suffer from "sensory deprivation," lacking the ability to incorporate visual evidence. Furthermore, concerns about memorization versus genuine discovery (Wu et al., 2024; Shojaee et al., 2025b) underscore the need for approaches that perform authentic data-driven reasoning rather than recalling known formulas.

By neglecting the crucial visual perceptual channel, existing methods are fundamentally constrained. They often resort to computationally expensive searches through vast equation spaces (Virgolin & Pissis, 2022), exhibit brittle token-matching behaviors, and fail to achieve the intuitive leaps that characterize human scientific breakthroughs. This limitation becomes particularly pronounced when dealing with complex dynamical systems where visual patterns in phase space and temporal evolution provide crucial insights that are difficult to extract from purely numerical data.

To bridge the gap between raw perception and abstract formalism, we introduce **VIPER-R1**, a **V**isual **I**nduction model for **P**hysics-based **E**quation **R**easoning. Rather than a mere pattern matcher, VIPER-R1 acts as a "computational phenomenologist," grounding symbolic reasoning in visual evidence by integrating plots, trajectory data, and symbolic logic to autonomously derive governing laws of motion.

Our framework draws inspiration from human scientific reasoning and follows a two-stage pipeline. In the first stage, **Motion Structure Induction (MSI)**, the model undergoes Supervised Fine-Tuning (SFT), learning to interpret kinematic evidence under joint supervision of Chain-of-Thought (CoT) rationales and ground-truth equations, before producing initial symbolic hypotheses guided by causal CoT prompts. In the second stage, **Reward-Guided Symbolic Calibration (RGSC)**, reinforcement learning with Group Relative Policy Optimization (GRPO) (Shao et al., 2024) refines these hypotheses using a structural reward function that favors topological correctness over coefficient matching. Finally, the model invokes an external symbolic regression tool for **Symbolic Residual Realignment (SR²)**, aligning theoretical expressions with empirical details to yield interpretable, precise formulas.

To support this research, we also release **PhysSymbol**, a large-scale corpus of 10,000 instances. Unlike prior datasets that treat discovery as pure regression, PhysSymbol provides a rich multimodal environment featuring comprehensive phase-space and field visualizations alongside expert-annotated Causal CoT traces.

Our contributions can be summarized as follows:

- We propose **VIPER-R1**, a multimodal framework that simulates the scientific reasoning process by deeply integrating visual perception with symbolic derivation.
- A hierarchical reasoning strategy is developed, consisting of Motion Structure Induction (MSI) for hypothesis generation, Reward-Guided Symbolic Calibration (RGSC) for refinement, and an agentic stage, Symbolic Residual Realignment (SR²), where external tools are employed to reconcile theoretical hypotheses with empirical data.
- We introduce **PhysSymbol**, a large-scale benchmark of 10,000 multimodal physics instances, created to advance research in vision-grounded scientific discovery.

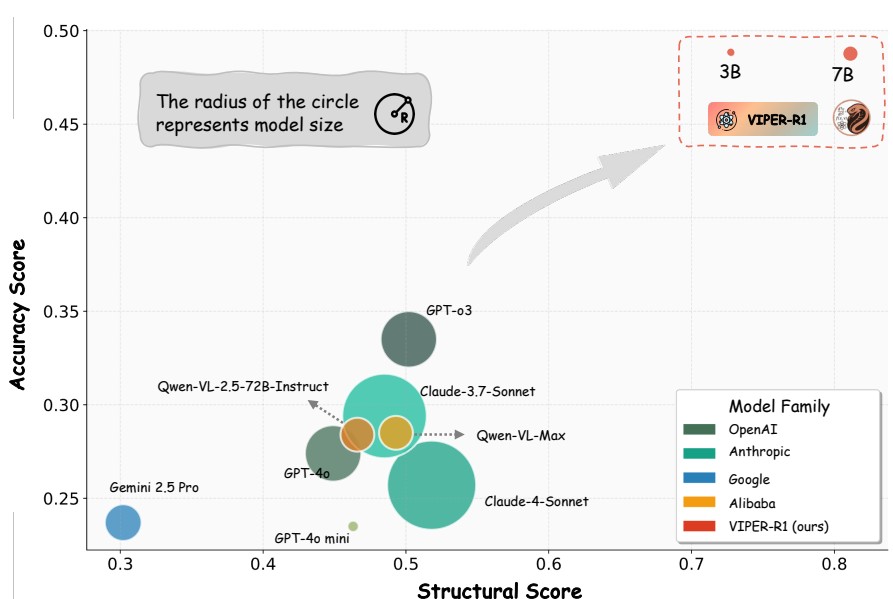

Figure 2: The performance of different SOTA Vision Language Models (VLMs) on physics formula discovery tasks.

## 2 RELATED WORK

### 2.1 SYMBOLIC REGRESSION FOR SCIENTIFIC DISCOVERY

Symbolic regression (SR) discovers mathematical expressions from data, evolving from genetic programming (Koza, 1994) to modern recursive (Udrescu & Tegmark, 2020) and evolutionary tools (Cranmer, 2023). Recent advances include Transformer-based mappings (Biggio et al., 2021; Kamienny et al., 2022) and hybrid systems integrating neural networks with RL (Petersen et al., 2019), MCTS (Sun et al., 2023), or guided genetic programming (Mundhenk et al., 2021; Meidani et al., 2023). However, SR remains NP-hard (Virgolin & Pissis, 2022; Shojaee et al., 2025a), often struggling with vast search spaces and physical plausibility (Virgolin & Pissis, 2022). Our work addresses this by employing a VLM to generate visually-grounded priors, refining blind search into targeted discovery.

### 2.2 LLMS FOR SCIENTIFIC DISCOVERY

LLMs have transformed automated science (Hu et al., 2025; Wang et al., 2023a; Lu et al., 2024), facilitating equation discovery via skeleton generation (Shojaee et al., 2025a), in-context learning (Merler et al., 2024), bilevel optimization (Ma et al., 2024a), and concept libraries (Grayeli et al., 2024). To mitigate memorization risks (Wu et al., 2024; Mirzadeh et al., 2024; Shojaee et al., 2025b), LLMs are also explored as optimization engines (Lehman et al., 2023; Romera-Paredes et al., 2024; Lange et al., 2024b) for prompts (Guo et al., 2023; Lange et al., 2024a) and neural architectures (Chen et al., 2023; Zheng et al., 2023a). Despite their success in hypothesis generation (Zheng et al., 2023b; Qi et al., 2023; Wang et al., 2023b; Majumder et al., 2024a; Li et al.,

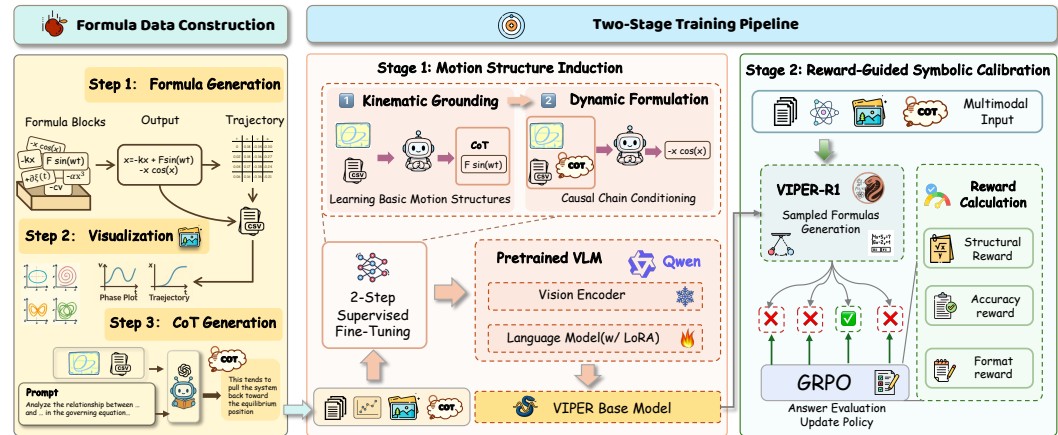

Figure 3: **Framework of VIPER-R1**. VIPER-R1 introduces a two-phase training framework for visual formula discovery and reasoning. First, a two-step curriculum, called Motion Structure Induction (MSI), is designed to imbue the VIPER-R1 with the ability to deduce the latent symbolic structure of a system's dynamics at stage 1. Subsequently, in stage 2, we employ reinforcement learning to "anneal" the model's generation policy, sharpening its focus on producing topologically correct physical laws.

2024; Wang et al., 2024; Ma et al., 2024b), the uni-modal nature of LLMs overlooks visual patterns essential to human scientific reasoning, a gap our work aims to bridge.

## 2.3 MULTIMODAL MODELS FOR SCIENTIFIC DISCOVERY

Vision Language Models (VLMs) increasingly support scientific tasks through visual reasoning (Zhang et al., 2024b; Su et al., 2025), from figure interpretation (Lu et al., 2022; Zhang et al., 2024a) to general understanding using models like GPT-4V (OpenAI, 2024), Qwen-VL (Bai et al., 2023), and Gemini (Google, 2025). While recent work derives equations from video coordinates (Li et al., 2025), we focus on direct visual reasoning from analytical plots (e.g., phase portraits) to hypothesize functional forms. Unlike existing benchmarks prone to memorization (La Cava et al., 2021; Matsubara et al., 2022; Majumder et al., 2024b; Wang et al., 2025; Shojaee et al., 2025b), our approach leverages a fine-tuned VLM for plot-based hypothesis generation, emulating the physicist's observation-reasoning cycle.

## 3 METHODOLOGY

Our proposed framework consists of a two-stage pipeline, as illustrated in Figure 3. The first stage involves two-step Motion Structure Induction with CoT reasoning. At stage 2, a RL-based refinement method is employed to help the model further calibrate the symbolic solution. When inferencing, a symbolic regression module is design as a optimal parameter searching tool to refine this hypothesis.

### 3.1 PROBLEM DEFINITION

The automated discovery of physical laws from multimodal empirical data can be formally defined as learning a mapping from a set of observations to the underlying symbolic law that governs the system. This process seeks to infer an interpretable symbolic expression $S$ from a diverse set of empirical evidence $\mathcal{E}$. The mapping can be represented as:

$$\pi_\theta : \mathcal{E} \to E,$$

where: $\mathcal{E} = \{\mathcal{V}, \mathcal{D}\}$ represents the complete set of Empirical Evidence, comprising both visual and numerical data modalities; $\mathcal{V} = \{V_1, V_2, \dots\}$ is a set of visual representations of the system's dynamics. For instance, in the context of the kinematic systems studied in this work, $\mathcal{V}$ typically includes a phase-space portrait ($I_{\text{phase}}$) and a time-series trajectory plot ($I_{\text{trajectory}}$); $\mathcal{D} = \{D_1, D_2, \dots\}$

is a set of quantitative measurements of the system's state variables. For the mechanical systems we investigate, $\mathcal{D}$ consists of time-series data of position, velocity, and acceleration, i.e., $\{(t_i, x(t_i), v(t_i), a(t_i))\}$; $E$ is the target output: an interpretable symbolic expression representing the governing physical law; $\pi_\theta$ is the parameterized model.

### 3.2 Motion Structure Induction (MSI)

The foundational stage of our framework is Motion Structure Induction (MSI), a specialized two-step curriculum designed to imbue the VIPER-R1 with the ability to deduce the latent symbolic structure of a system's dynamics from its visual phenomenological representations. This process explicitly emulates the cognitive progression from qualitative observation to quantitative hypothesis.

#### 3.2.1 Step 1: Joint Induction of Causal Reasoning and Symbolic Structure

The initial stage mirrors a physicist's first encounter with a new phenomenon: concurrently observing, reasoning, and formulating a preliminary idea. Here, the VIPER-R1 is trained to jointly generate both a Causal Chain of Thought (C-CoT) and an initial Symbolic Ansatz. The input is the complete set of Empirical Evidence $E = (\mathcal{V}, \mathcal{D})$. The model's objective is to maximize the likelihood of the entire structured output, which comprises the reasoning chain $C$ followed by the symbolic law $S$.

This joint objective is crucial; it compels the VIPER-R1 to ground its symbolic output in an explicit, physically-motivated reasoning process. The model must learn not just *what* the governing law is, but *why* it takes that form, based on visual cues within the evidence. Formally, we define the training objective for this stage by maximizing the log-probability of the target sequence $Y = (C, S)$:

$$\mathcal{L}_{\text{MSI-1}} = -\mathbb{E}_{(E,Y)\sim\mathcal{D}_{\text{phys}}} \sum_{t=1}^{|Y|} \log \pi_\theta(y_t \mid E, y_{<t}), \tag{1}$$

where $\mathcal{D}$ is our PhysSymbol Corpus, $Y = (y_1, ..., y_{|Y|})$ is the concatenated sequence of the C-CoT and the symbolic law, and $\pi_\theta$ is the policy of the VIPER-R1.

#### 3.2.2 Step 2: C-CoT-Guided Symbolic Formulation

The second stage of our curriculum refines the VIPER-R1's ability to translate a well-formed physical argument into a precise symbolic form. This is analogous to a physicist taking their detailed notes and meticulously composing the final equation. In this stage, the model is provided with both the empirical evidence $E$ and the ground-truth C-CoT, $C$, and is tasked *only* with generating the correct symbolic law $S$.

By conditioning on an ideal reasoning chain, we allow the model to dedicate its full representational capacity to mastering the complex syntax and semantics of physical formalisms. This decouples the task of reasoning from the task of formulation. The loss is computed exclusively on the tokens of the symbolic law $S$:

$$\mathcal{L}_{\text{MSI-2}} = -\mathbb{E}_{(E,C,S)\sim\mathcal{D}_{\text{phys}}} \sum_{t=1}^{|S|} \log \pi_\theta(s_t \mid E, C, s_{<t}). \tag{2}$$

The two-stage MSI curriculum is designed with two key considerations. First, by decoupling the complex cognitive task of causal reasoning from the intricate syntactic task of symbolic formulation, it enhances both the stability and the effectiveness of learning. Second, this curriculum reflects a hierarchical abstraction process: it encourages the model to first construct a high-level qualitative understanding through C-CoT, and only then proceed to generate a low-level, precise symbolic output—thereby mirroring effective human problem-solving strategies.

Upon completion of MSI, the resulting model, $\pi_{\text{VIPER}}$, possesses a robust, physically-grounded foundation, ready for the subsequent Reward-Guided Symbolic Calibration stage.

### 3.3 Reward-Guided Symbolic Calibration (RGSC)

Following the foundational MSI phase, the VIPER-R1 possesses the ability to generate plausible symbolic hypotheses. However, to further enhance the structural purity and reliability of these

hypotheses, we introduce a refinement phase: Reward-Guided Symbolic Calibration (RGSC). This stage employs reinforcement learning to "anneal" the model's generation policy, sharpening its focus on producing topologically correct physical laws. We select the Group Relative Policy Optimization (GRPO) algorithm (Shao et al., 2024) for this task, as it is highly efficient for large-scale models and circumvents the need for a separate, computationally expensive value network. GRPO's design, which computes advantages relative to a batch of sampled actions, is exceptionally well-suited for our task where a direct, analytical reward can be computed for any generated symbolic expression.

**Sampling a Distribution of Symbolic Hypotheses.** For each instance of Empirical Evidence $E = (\mathcal{V}, \mathcal{D})$ from our PhysSymbol Corpus, we sample a group of $G$ candidate symbolic expressions $\{S_1, S_2, \ldots, S_G\}$ from the current policy $\pi_\theta$, which is initialized from the model fine-tuned during MSI. This sampling process is defined as:

$$S_i \sim \pi_\theta(S \mid E), \quad \text{for } i = 1, 2, \ldots, G. \tag{3}$$

This strategy encourages exploration within the vast space of possible physical theories, allowing the model to discover and reinforce more robust and accurate symbolic structures.

**Formulating the Structural Reward.** Each sampled ansatz $S_i$ is evaluated and assigned a reward $R(S_i)$. Our reward function is designed to align with the central goal of discovering structurally correct physical laws, regardless of specific coefficient values. It consists of three weighted components: a Format Reward ($R_{\text{format}}$), our novel Parameter-Agnostic Structural Reward ($R_{\text{structural}}$), and an Exact Match Accuracy Reward ($R_{\text{accuracy}}$).

$$R(S_i) = w_f R_{\text{format}}(S_i) + w_s R_{\text{structural}}(S_i, S_{\text{GT}}) + w_a R_{\text{accuracy}}(S_i, S_{\text{GT}}). \tag{4}$$

• **Format Reward ($R_{\text{format}}$):** This binary reward component ensures the model's output adheres strictly to the predefined `<think>`...`<answer>` template, which is crucial for interpretability and reliable parsing. It awards 1 for correct formatting and 0 otherwise.

• **Parameter-Agnostic Structural Reward ($R_{\text{structural}}$):** This is the core of our reward mechanism, evaluating the fundamental correctness of the generated law's structure. As detailed in Appendix B.3, it calculates the Jaccard similarity between the "structural skeletons" of the generated ansatz and the canonical equation. This metric rewards topological correctness over superficial coefficient matching, aligning the optimization objective with the VIPER-R1's primary role.

• **Exact Match Accuracy Reward ($R_{\text{accuracy}}$):** This component provides the strictest evaluation, awarding a binary reward of 1 only if the generated formula $S_i$ is symbolically identical to the ground truth $S_{\text{GT}}$. This encourages ultimate precision of the model's output.

**Policy Update with Relative Advantage.** The rewards $\{r_1, r_2, \ldots, r_G\}$ for the group of sampled hypotheses are normalized to compute their relative advantages, preventing instability from high-variance rewards. The relative advantage $A_i$ for each ansatz $S_i$ is defined as:

$$A_i = \frac{r_i - \text{mean}(r_1, \ldots, r_G)}{\text{std}(r_1, \ldots, r_G) + \epsilon}. \tag{5}$$

The policy $\pi_\theta$ is then updated to increase the likelihood of generating hypotheses with positive advantages. This process is further regularized by a Kullback–Leibler divergence penalty between the updated policy and the original reference policy from the MSI stage, ensuring stable learning and preventing the model from deviating too far from its physically-grounded foundation.

## 3.4 AGENTIC REFINEMENT VIA SYMBOLIC RESIDUAL REALIGNMENT (SR²)

Upon completing its internal calibration, the VIPER-R1 has produced a high-confidence Symbolic Ansatz, denoted as $S_0$. This expression represents a robust, first-order approximation of the system's dynamics. In the final stage of our framework, the VIPER-R1 transitions into an *agentic* role. It recognizes that while its ansatz predicts a target variable $\hat{a}_{\text{VLM}}$, a discrepancy or "residual field" may exist between this theoretical model and the precise empirical evidence.

To characterize and correct for this residual, the VIPER-R1 agentically invokes an external tool: a high-performance symbolic regression engine (Cranmer, 2023). We term this sophisticated tool-use process **Symbolic Residual Realignment (SR²)**. This technique mirrors a physicist performing a perturbation analysis to account for higher-order effects, thereby realigning their theory with empirical reality.

**The SR² Process.** The core principle of SR² is to dramatically simplify the task for the symbolic regression tool. Instead of tasking it with searching the entire, near-infinite space of possible physical laws, we constrain its search to the much smaller, well-behaved space of the residual error. The process unfolds as follows:

**Step 1: Residual Field Calculation:** The residual field, $r(t)$, is computed as the difference between the ground-truth target values from the empirical data, $a_{\mathrm{GT}}(t)$, and the prediction from the VIPER-R1's Symbolic Ansatz:

$$r(t) = a_{\mathrm{GT}}(t) - \hat{a}_{\mathrm{VLM}}(x, v, t). \tag{6}$$

**Step 2: Symbolic Regression on the Residual:** The SR engine is then deployed with the explicit goal of finding a parsimonious and accurate symbolic expression, $S_{\mathrm{residual}}$, that best models the residual field $r(t)$. This focused task allows the SR tool to operate with maximum efficiency.

$$a_{\mathrm{residual}}(x, v, t) \leftarrow \mathrm{SR}(x, v, t, r(t)). \tag{7}$$

**Step 3: Theory Realignment:** The final, empirically-realigned Law of Motion, $S_{\mathrm{final}}$, is constructed by composing the VIPER-R1's initial ansatz with the discovered residual expression. This yields a complete and highly accurate model of the system's dynamics.

$$a_{\mathrm{final}}(x, v, t) = \hat{a}_{\mathrm{VLM}}(x, v, t) + a_{\mathrm{residual}}(x, v, t). \tag{8}$$

The whole process of SR² is summarized in Algorithm 2. More information is described in Appendix D.

## 3.5 THE PHYSSYMBOL DATASET

To support this research, we introduce PHYSSYMBOL, a multimodal dataset bridging visual phenomenology and symbolic expressions. Unlike numerical-only benchmarks (Shojaee et al., 2025b), PHYSSYMBOL incorporates visual intuition through two key features:

- **Dual-Visualization:** Combines time-domain ($x$-$t$) and phase-space ($v$-$x$) plots, enabling models to correlate temporal patterns with geometric stability signatures (e.g., limit cycles).

- **Causal CoT Annotations:** Expert-verified reasoning traces that explicitly map visual features (e.g., amplitude decay) to symbolic terms (e.g., $-cv$).

The dataset comprises 10,000 instances split into Training (Mechanics) and Research (Cross-disciplinary) subsets, as detailed in Table 1.

Table 1: Statistics of the PhysSymbol Dataset

| Feature | Part 1: Mechanics (Training) | Part 2: Research (OOD) |
|---|---|---|
| Total Instances | 5,000 | 5,000 |
| Physics Domains | Classical Mechanics | EM, Fluid, Thermo, Quantum |
| Equation Types | ODEs (2–5 terms) | PDEs (Field Equations) |
| Visual Modalities | Trajectory & Phase Plots | Scalar/Vector Fields & Gradients |
| Reasoning Labels | Causal CoT Traces | Ground Truth Formulas |

# 4 EXPERIMENTS

## 4.1 EXPERIMENTAL SETUP

**Dataset.** To avoid data contamination, where models might have seen common benchmarks during pre-training, we adopt the strategy of constructing a new synthetic dataset, as advocated in prior work such as LLM-SR (Shojaee et al., 2025a). All experiments are performed on our purpose-built **PhysSymbol** corpus, where each instance represents a distinct and complex physical system. Further details on data generation and statistics are provided in Appendix D.

**Models and Baselines.** Our primary models, **VIPER-R1-3B** and **VIPER-R1-7B**, are based on the Qwen-VL-2.5 3B and 7B architectures, respectively. We compare against a SR-based method PySR (Cranmer, 2023), and a diverse set of state-of-the-art MLMs, including *GPT-5* (OpenAI, 2025a), *GPT-5 mini* (OpenAI, 2025a), *GPT-4o mini* (OpenAI, 2025b), *GPT-4o* (OpenAI, 2024), *Grok 3* (xAI, 2025), *GPT-o3* (OpenAI, 2025b), *Claude-4 Sonnet* (Anthropic, 2024), *Claude-3.7 Sonnet* (Anthropic, 2025), *Qwen-VL-Max* (Bai et al., 2023), *Qwen-VL-2.5-72B-Instruct* (Bai et al., 2025), and *Gemini 2.5 Pro* (Google, 2025).

**Evaluation Metrics.** We evaluate the models across several dimensions to capture different aspects of performance:

- **Structural Score ($S_{\text{struct}}$):** This is our primary metric for the VLM's hypothesis generation capability. It is the parameter-agnostic Jaccard similarity between the terms of the generated formula and the canonical equation. A score of 1.0 indicates a perfect structural match.

- **Accuracy Score ($S_{\text{acc}}$):** A stricter metric that measures the rate of exact symbolic matches between the generated formula and the canonical equation.

- **Post-SR² MSE:** The final Mean Squared Error of the complete, realigned formula after the **SR²** stage. This measures the end-to-end performance of the entire framework. A lower value is better.

Further experimental details, including model architectures, training procedures, and evaluation protocols, are provided in Appendix B.

## 4.2 MAIN RESULTS AND ANALYSIS

To ensure statistical robustness, we benchmarked VIPER-R1 against a comprehensive suite of SOTA VLMs and traditional SR methods. We report the mean $\pm$ 95% confidence intervals across 5 independent runs. The results in Table 2 demonstrate our framework's significant superiority.

Table 2: Main results comparing VIPER-R1 against SOTA VLMs and SR-based methods on the PhysSymbol test set. We report the mean $\pm$ 95% confidence interval. Our method achieves the highest structural and accuracy scores.

| Category | Method | Structural Score ($S_{\text{struct}}$) ↑ | Accuracy Score ($S_{\text{acc}}$) ↑ | Post-SR² MSE ↓ |
|---|---|---|---|---|
| VLMs | GPT-5 | $0.494 \pm 0.012$ | $0.363 \pm 0.015$ | $0.192 \pm 0.018$ |
| | GPT-5-mini | $0.455 \pm 0.010$ | $0.350 \pm 0.011$ | $0.154 \pm 0.013$ |
| | GPT-4o mini | $0.463 \pm 0.014$ | $0.235 \pm 0.009$ | $0.109 \pm 0.012$ |
| | GPT-4o | $0.449 \pm 0.011$ | $0.274 \pm 0.013$ | $0.286 \pm 0.025$ |
| | Grok 3 | $0.026 \pm 0.005$ | $0.019 \pm 0.003$ | $0.177 \pm 0.020$ |
| | GPT-o3 | $0.502 \pm 0.015$ | $0.335 \pm 0.014$ | $0.234 \pm 0.019$ |
| | Claude-4-Sonnet | $0.518 \pm 0.009$ | $0.257 \pm 0.010$ | $0.091 \pm 0.008$ |
| | Claude-3.7-Sonnet | $0.485 \pm 0.013$ | $0.294 \pm 0.012$ | $0.136 \pm 0.015$ |
| | Qwen-VL-Max | $0.493 \pm 0.016$ | $0.285 \pm 0.014$ | $0.210 \pm 0.022$ |
| | Qwen-VL-2.5-72B | $0.466 \pm 0.014$ | $0.284 \pm 0.015$ | $0.198 \pm 0.017$ |
| | Gemini 2.5 Pro | $0.302 \pm 0.018$ | $0.237 \pm 0.016$ | $0.107 \pm 0.011$ |
| SR-Only | PySR | $0.352 \pm 0.021$ | $0.048 \pm 0.025$ | $\mathbf{5.6e^{-3} \pm 1.2e^{-4}}$ |
| Ours | VIPER-R1-3B | $0.728 \pm 0.006$ | $0.488 \pm 0.007$ | $0.081 \pm 0.005$ |
| | VIPER-R1-7B | $\mathbf{0.812 \pm 0.004}$ | $\mathbf{0.487 \pm 0.005}$ | $0.032 \pm 0.003$ |

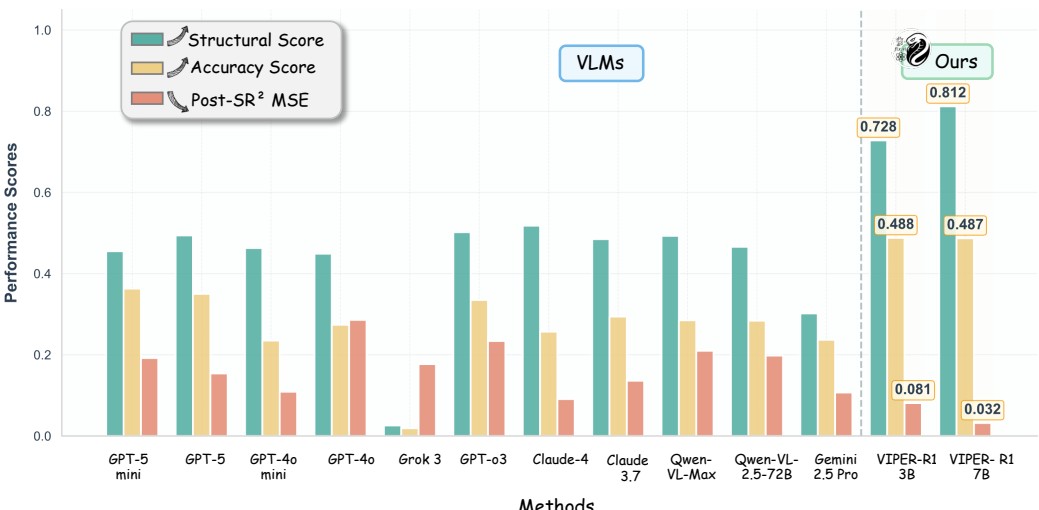

Figure 4: Quantitative comparison of model performance on the PhysSymbol test set. We report three metrics: structural score , accuracy score, and post-symbolic-regression MSE. VIPER-R1 (ours) outperforms all VLM baselines across all metrics, demonstrating significant improvements in both symbolic structure induction and predictive accuracy.

**Superiority in Initial Hypothesis Generation.** The Structural ($S_{\text{struct}}$) and Accuracy ($S_{\text{acc}}$) scores evaluate the quality of the generated formula structure. Our VIPER-R1-7B achieves a dominant $S_{\text{struct}}$ of $0.812$, representing a $56.7\%$ improvement over the best VLM baseline (Claude-4-Sonnet). Notably, while the traditional SR method (PySR) achieves low error via numerical fitting, it fails to identify the correct physical structure ($S_{\text{struct}} = 0.352$), highlighting the necessity of visual grounding. Our two-stage training curriculum successfully imbues the model with this domain-specific, physicist-like intuition.

**Excellence in Final Law Discovery.** A high-quality initial hypothesis is critical for finding the true global optimum. While SR methods prone to overfitting, like PySR, achieve extremely low MSE, they often miss the true physical law. VIPER-R1 strikes the optimal balance: by providing a structurally correct ansatz, it enables the solver to reach a state-of-the-art MSE of $0.032$ among interpretable models, nearly $3\times$ lower than the best VLM baseline ($0.091$). Even our 3B model significantly outperforms larger general-purpose models.

## 5 CONCLUSION

In this work, we introduced the **VIPER-R1**, a **V**isual **I**nduction model for **P**hysics-based **E**quation **R**easoning, which emulates the scientific workflow by grounding symbolic reasoning in visual perception. Through a carefully designed curriculum of Motion Structure Induction and Reward-Guided Symbolic Calibration, we trained a "computational phenomenologist" capable of generating high-quality symbolic hypotheses from visual data. The subsequent agentic deployment of a Symbolic Residual Realignment stage demonstrated how this VLM use external tools to refine its own theories, achieving remarkable precision. Experiments on the PhysSymbol benchmark demonstrate that this multimodal, multi-stage approach significantly outperforms state-of-the-art VLMs. In the future, our approach invites further exploration. Scaling to larger and more diverse datasets, including chaotic systems and PDEs, promises even stronger performance. Moving from simulated plots to real experimental videos would expose the model to the full complexity of empirical observation.

**Ethics Statement.** All authors have read and comply with the ICLR Code of Ethics. This work involves no human subjects or sensitive data, and we are unaware of any potential misuse, harm, or bias. No conflicts of interest or compromising sponsorships exist.

**Reproducibility Statement.** We have made extensive efforts to ensure the reproducibility of our results. Details of the proposed methodology, training procedure, hyperparameters, and evaluation metrics are provided in Sections 3 and 4. We include complete algorithm pseudocode in Appendix B.4 and provide ablation studies in Appendix C.3. A full description of the datasets and preprocessing steps is provided in Appendix D.

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

# A  APPENDIX

This supplementary material provides additional details on the proposed method and experimental results that could not be included in the main manuscript due to page limitations. Specifically, this appendix is organized as follows.

- Sec. B describes the training setup, model configurations, and additional evaluation details, offering comprehensive experimental specifications.
- Sec. C presents supplementary experiments, including ablation studies and comparisons with other fine-tuned baselines.
- Sec. D provides more details about PhysSymbol and discusses how we collected, filtered, and reconstructed a high-quality dataset.
- Sec. E includes more visualization cases.

# B  DETAILS OF TRAINING AND EVALUATION

## B.1  TRAINING SETTINGS AND DETAILS

This subsection provides complete implementation details for the three core components of VIPER-R1: MSI, RGSC, and Symbolic Residual Realignment.

**MSI Training Configuration.** MSI fine-tunes the base VLM using supervised learning to induce visually grounded symbolic structures. Training employs the AdamW optimizer with a learning rate of $2 \times 10^{-5}$, cosine annealing, and a warmup duration of the first 300 optimization steps. A batch size of 64 is used throughout, with gradient clipping at 1.0 and weight decay set to 0.1. The model processes sequences up to 4096 tokens and is trained for three epochs over PhysSymbol. The total loss combines three components: an equation-token loss that teaches the symbolic structure, a causal CoT generation loss that preserves interpretable reasoning, and a visual-alignment loss that encourages consistency between plot features and symbolic predictions. These losses are mixed with weights 1.0, 0.3, and 0.1, respectively. To enhance stability for long symbolic sequences, teacher forcing is applied to approximately 40% of C-CoT steps, which substantially reduces exposure bias in symbolic reasoning.

Table 3: MSI supervised fine-tuning hyperparameters.

| Hyperparameter | Setting |
|---|---|
| Optimizer | AdamW |
| Learning rate | $2 \times 10^{-5}$ |
| Warmup steps | 300 |
| Batch size | 64 |
| Gradient clipping | 1.0 |
| Weight decay | 0.1 |
| Max sequence length | 4096 tokens |
| Epochs | 3 |
| Loss weights | $L_{\text{eq}} : L_{\text{cot}} : L_{\text{vis}} = 1.0 : 0.3 : 0.1$ |
| Teacher forcing | 40% of C-CoT steps |

**Reward-Guided Symbolic Calibration (RGSC).** RGSC refines the symbolic hypothesis through a GRPO-based reinforcement learning procedure tailored for equation structure purification. Each training batch contains 32 prompts, with the model sampling six candidate symbolic expressions per instance, yielding an effective batch of 192 hypotheses. The optimizer remains AdamW, operating at a learning rate of $5 \times 10^{-6}$, while advantage estimates are normalized per batch via a z-score transform to stabilize policy gradients. A KL penalty of $\beta = 0.02$ ensures that symbolic updates remain faithful to the pretrained distribution unless reward signals strongly encourage structural deviation.

The reward used in RGSC is a weighted sum of three components:

$$R = w_s R_{\text{struct}} + w_f R_{\text{func}} + w_a R_{\text{ans}}.$$

The structural reward $R_{\text{struct}}$ measures the symbolic alignment between the predicted operator set and the ground-truth operator set via a Jaccard similarity:

$$R_{\text{struct}} = \text{Jaccard}(\mathcal{O}_{\text{pred}}, \mathcal{O}_{\text{gt}}),$$

where $\mathcal{O}$ includes operators such as $+, -, \sin, \cos, x, v, t$, and nonlinear terms. In practice, $w_s = 5$ dominates the reward landscape, while $w_f = 1$ enforces functional plausibility and $w_a = 1$ preserves syntactic correctness in the final generated expressions.

Table 4: RGSC reinforcement learning hyperparameters.

| Hyperparameter | Setting |
|---|---|
| Rollout length | 4 |
| Samples per instance | 6 |
| Batch size | 32 (192 hypotheses) |
| Learning rate | $5 \times 10^{-6}$ |
| Optimizer | AdamW |
| KL coefficient $\beta$ | 0.02 |
| Advantage normalization | Per-batch z-score |
| Reward weights | $w_s = 5, \ w_f = 1, \ w_a = 1$ |

**SR$^2$ (Symbolic Residual Realignment).** The SR$^2$ module corrects systematic deviations between the MSI+RGSC symbolic expression and the true dynamics by fitting the residual acceleration

$$r(t) = a(t) - \hat{S}_{\text{MSI+RGSC}}(x(t), v(t), t)$$

using symbolic regression. PySR serves as the primary backend, with a population size of 100, 50 000 search iterations, and an operator library containing arithmetic, trigonometric, and exponential primitives. The maximum expression depth is restricted to 6, and a small complexity penalty $\lambda = 0.001$ is applied to discourage overly expressive corrections.

**Training Compute.** All training runs were performed on NVIDIA A800 (80GB) GPUs. As shown in Table 5, MSI requires only a short supervised phase, while RGSC, our reinforcement-learning based symbolic calibration stage—constitutes the majority of the compute due to multi-sample rollout generation and reward evaluation. Despite this, the overall training cost remains modest compared to typical large-model finetuning workloads, as VIPER-R1 focuses on targeted structural refinement rather than end-to-end large-scale retraining.

Table 5: Training Computational Cost Analysis

| Component | Hardware | Duration | Total Cost (GPU-hours) |
|---|---|---|---|
| MSI | $8 \times$ A800 | 4 hours | 32 |
| RGSC | $8 \times$ A800 | 42 hours | 336 |
| **Total** | - | - | **368** |

## B.2 SYSTEM PROMPTS

The behavior and reasoning process of VIPER-R1 are carefully guided by a series of structured system prompts tailored to each stage of our training and inference pipeline. These prompts define the model's role as a scientific assistant and establish the expected format for its reasoning and final output. This structured approach is crucial for decoupling complex tasks and progressively building the model's capability for scientific discovery. Below, we detail the specific prompts used in each phase.

### B.2.1 PROMPT FOR MSI STEP 1

In the initial MSI step, as shown in Figure 5, the goal is to teach the model to perform end-to-end reasoning, connecting raw visual phenomena directly to a final governing equation. The prompt

instructs the model to act as a scientific assistant, verbalize its step-by-step analysis, and provide a conclusive answer in a structured format.

---

**System Prompt**

```
You are a helpful scientific assistant. Given trajectory
images and motion data from a physical system, reason
step-by-step to explain the observed behavior, then
output the governing equation. Wrap your reasoning
process in <think> </think> and your final equation
in <answer> </answer>.
```

---

Figure 5: System prompt for the first SFT stage (MSI).

### B.2.2  PROMPT FOR MSI STEP 2

In the second step of MSI, as shown in Figure 6, we decouple the task: the model is provided with the pre-computed reasoning chain (C-CoT) and is tasked only with translating this analysis into a precise symbolic equation. This prompt focuses the model's training on the final, crucial step of symbolic formulation.

---

**System Prompt**

```
You are a helpful scientific assistant. Given the
reasoning steps for a physical system and its
trajectory images, output the corresponding governing
equation. The reasoning is provided in <think> </think>
tags, and your answer should be placed in
<answer> </answer> tags.
```

---

Figure 6: System prompt for the second SFT stage (CoT-Aware).

### B.2.3  PROMPT FOR RGSC

During the reinforcement learning phase, as shown in Figure 7, the prompt is refined to encourage more abstract and generalized symbolic reasoning. It explicitly asks the model to use symbolic placeholders for unknown parameters, which is essential for discovering general physical laws rather than fitting to specific numerical instances. This prompt guides the generation of hypotheses that are then evaluated by our reward function.

---

**System Prompt**

```
The user provides visual and trajectory data of a
physical phenomenon. The Assistant's task is to act
as a physicist. First, think step-by-step about the
underlying physical principles in <think> tags. Then,
derive and state the final governing equation in
<answer> tags. The equation should use symbolic
placeholders for unknown parameters (e.g., k, c, F)
and standard variables for the system (x, v, t).
```

---

Figure 7: System prompt for the RGSC stage.

## B.3 Detailed Reward Function Formulation for RGSC

The total reward signal $R(S_i)$ used during the Reward-Guided Symbolic Calibration (RGSC) stage is a weighted sum of three distinct components. Each component is designed to evaluate a specific aspect of the generated Symbolic Ansatz $S_i$, allowing for a balanced and effective policy optimization. The composite reward is defined as:

$$R(S_i) = w_f R_{\text{format}}(S_i) + w_s R_{\text{structural}}(S_i, S_{\text{GT}}) + w_a R_{\text{accuracy}}(S_i, S_{\text{GT}}), \tag{9}$$

where $S_{\text{GT}}$ is the Canonical Governing Equation, and $w_f, w_s, w_a$ are hyperparameters that weight the contribution of each reward component. Below, we detail the formulation of each component.

**Format Reward ($R_{\text{format}}$).**   The primary purpose of this reward is to enforce a consistent and parsable output structure, which is crucial for both interpretability and automated evaluation. We use regular expressions to verify that the model's output strictly adheres to our predefined template, which requires a reasoning process enclosed within `<think>...</think>` tags followed by a final symbolic formula within `<answer>...</answer>` tags. This is a binary reward:

$$R_{\text{format}}(S_i) = \begin{cases} 1 & \text{if format is correct} \\ 0 & \text{otherwise} \end{cases} \tag{10}$$

**Parameter-Agnostic Structural Reward ($R_{\text{structural}}$).**   This is the most critical component for our task, designed to assess the fundamental topological correctness of the posited physical law. It rewards the model for identifying the correct basis functions and their relationships (e.g., $-k * x$, $-c * v$), irrespective of the specific values or symbols used for the coefficients (e.g., $k$, $c$). The calculation involves two steps:

1. Both the generated ansatz $S_i$ and the ground truth $S_{\text{GT}}$ are parsed into symbolic expressions. We then decompose each expression into a set of constituent terms. For additive expressions, these are the terms separated by addition. For non-additive expressions, the term is the expression itself.

2. Each term is then normalized into a "structural skeleton" by replacing all numerical coefficients and symbolic parameters with a signed unit (i.e., $+1$ or $-1$), while preserving the core physical variables ($x, v, t$) and mathematical operators.

The final reward is the Jaccard similarity between the set of skeletonized terms from the generated formula ($\mathcal{T}_{\text{gen}}$) and the ground truth ($\mathcal{T}_{\text{GT}}$). This provides a fine-grained score between 0 and 1.

$$R_{\text{structural}}(S_i, S_{\text{GT}}) = \frac{|\mathcal{T}_{\text{gen}} \cap \mathcal{T}_{\text{GT}}|}{|\mathcal{T}_{\text{gen}} \cup \mathcal{T}_{\text{GT}}|} \tag{11}$$

**Exact Match Accuracy Reward ($R_{\text{accuracy}}$).**   This component provides the strictest evaluation, rewarding the model only if its generated symbolic formula is mathematically identical to the ground truth. This encourages ultimate precision, especially for formulas where parameters are also represented symbolically (e.g., using 'k' instead of a number). We leverage the `sympy` library to perform a robust symbolic comparison. Both the generated answer and the ground truth are parsed into symbolic expressions, and we check if their simplified difference is zero. This is also a binary reward:

$$R_{\text{accuracy}}(S_i, S_{\text{GT}}) = \begin{cases} 1 & \text{if } \texttt{sympy.simplify}(S_i - S_{\text{GT}}) = 0 \\ 0 & \text{otherwise} \end{cases} \tag{12}$$

By combining these three reward signals, we create a rich and nuanced optimization landscape. The model is primarily guided by the structural reward ($w_s$ is typically the largest weight) to learn the correct physics, while also being encouraged to produce well-formatted and, when possible, exactly correct symbolic expressions.

## B.4 Algorithm Pseudocode

To provide a comprehensive and reproducible overview of our methodology, we present the detailed pseudocode for our framework's two primary components: the end-to-end training process and the inference-time refinement procedure.

Algorithm 1 outlines the complete training framework for the VIPER-R1. This algorithm details the two primary phases through which the model is forged: first, the supervised Motion Structure Induction curriculum, which teaches the model to form hypotheses from visual data; and second, the subsequent reinforcement learning phase of Reward-Guided Symbolic Calibration, which purifies the model's symbolic generation policy.

Following this, Algorithm 2 describes the inference procedure. This algorithm formalizes the Agentic Refinement via Symbolic Residual Realignment process, wherein the fully trained VIPER-R1 generates an initial hypothesis and then proactively invokes an external symbolic regression tool to produce a final, empirically-realigned physical law.

---

**Algorithm 1:** VIPER-R1 Training Framework: MSI and RGSC

---

**Inputs** : The PhysSymbol Corpus $\mathcal{D}$,
        Initial model parameters $\theta_0$ from a pre-trained VLM,
        Number of MSI steps $N_{\text{MSI-1}}, N_{\text{MSI-2}}$,
        Number of RGSC steps $N_{\text{RGSC}}$,
        GRPO group size $G$,
        Reward weights $w_f, w_s, w_a$

**Outputs:** The final, calibrated VIPER-R1 policy $\pi_{\text{RGSC}}$

---

// Phase 1: Motion Structure Induction (MSI)

1   $\pi_\theta \leftarrow$ InitializeModel($\theta_0$);

   // Step 1.1: Joint Induction of C-CoT and Symbolic Structure

2   **for** $i = 1$ *to* $N_{MSI\text{-}1}$ **do**

3       Sample a batch $(E, Y) \sim \mathcal{D}$, where $Y = (C, S)$;

4       Update $\theta$ by descending the gradient of $\mathcal{L}_{\text{MSI-1}}$ w.r.t. Eq. equation 1;

5   **end**

   // Step 1.2: C-CoT-Guided Symbolic Formulation

6   **for** $i = 1$ *to* $N_{MSI\text{-}2}$ **do**

7       Sample a batch $(E, C, S) \sim \mathcal{D}$;

8       Update $\theta$ by descending the gradient of $\mathcal{L}_{\text{MSI-2}}$ w.r.t. Eq. equation 2;

9   **end**

10   $\pi_{\text{MSI}} \leftarrow \pi_\theta$

---

// Phase 2: Reward-Guided Symbolic Calibration (RGSC)

11   $\pi_\theta \leftarrow \pi_{\text{MSI}}$; $\pi_{\text{ref}} \leftarrow \pi_{\text{MSI}}$

12   **for** $k = 1$ *to* $N_{RGSC}$ **do**

13       Sample a batch of Empirical Evidence $E \sim \mathcal{D}$;

14       **for** $j = 1$ *to* $G$ **do**

15          $S_j \sim \pi_\theta(S \mid E)$;

16       **end**

17       **for** $j = 1$ *to* $G$ **do**

18          $r_j \leftarrow w_f R_{\text{format}}(S_j) + w_s R_{\text{structural}}(S_j, S_{\text{GT}}) + w_a R_{\text{accuracy}}(S_j, S_{\text{GT}})$;

19       **end**

20       $A \leftarrow$ Normalize($r_1, \ldots, r_G$);

21       Update $\theta$ using advantages $A$ and a KL penalty against $\pi_{\text{ref}}$;

22   **end**

23   $\pi_{\text{RGSC}} \leftarrow \pi_\theta$;

24   **return** $\pi_{RGSC}$;

---

### B.5 EVALUATION METRICS

To provide a holistic assessment of our framework's performance, we employ a suite of distinct metrics, each designed to capture a different facet of success, from high-level structural correctness to final empirical accuracy.

**Structural Score** ($S_{\text{struct}}$). This is our primary metric for evaluating the core capability of the VIPER-R1: its ability to generate a topologically correct symbolic hypothesis. This score measures

---

**Algorithm 2:** Agentic Refinement via Symbolic Residual Realignment (SR²)

---

**Inputs** : Trained VIPER-R1 policy $\pi_{\text{VIPER-R1}}$,
           Empirical Evidence $E = (\mathcal{I}, \mathcal{D})$,
           Symbolic Regression engine $\mathcal{SR}$
**Outputs:** The final, realigned Law of Motion $S_{\text{final}}$

---

   // Stage 1: VLM Hypothesis Generation
1  $S_0 \leftarrow \text{GenerateAnsatz}(\pi_{\text{VIPER-R1}}, E)$;
2  $a_{\text{VLM}} \leftarrow \text{CompileFunction}(S_0)$
   // Stage 2: Residual Field Calculation
3  $a_{\text{GT}}(x, v, t) \leftarrow \text{ExtractData}(E.\mathcal{D}) \; r \leftarrow a_{\text{GT}} - \hat{a}_{\text{VLM}}(x, v, t)$
   // Stage 3: Tool-Using for Residual Modeling
4  $S_{\text{residual}} \leftarrow \mathcal{SR}(\text{inputs} = (x, v, t), \text{target} = r)$
   // Stage 4: Theory Realignment
5  $S_{\text{final}} \leftarrow S_0 + S_{\text{residual}}$
6  **return** $S_{final}$;

---

the structural similarity between the generated formula and the canonical equation, intentionally ignoring numerical coefficients to focus purely on the underlying physical structure.

**Accuracy Score ($S_{\text{acc}}$).** To measure the exactness of the generated formulas, we use a strict symbolic accuracy score. This metric evaluates whether the generated formula is mathematically identical to the canonical equation. It serves as a challenging measure of the model's ultimate precision.

**Post-SR² Mean Squared Error (MSE).** This metric evaluates the end-to-end performance of the entire VIPER-R1 framework by measuring how well the final, refined formula fits the observed data. It quantifies the empirical accuracy after the **SR²** stage has been completed.

*Calculation:* Let $S_{\text{final}}$ be the final symbolic law produced by our framework. This expression is converted into a callable function $a_{\text{final}}(x, v, t)$. The MSE is then computed over the $N$ data points in the test set's trajectory data:

$$\text{MSE} = \frac{1}{N} \sum_{i=1}^{N} \left( a_{\text{GT}}(t_i) - a_{\text{final}}(x_i, v_i, t_i) \right)^2, \tag{13}$$

where $a_{\text{GT}}(t_i)$ is the ground-truth acceleration at time $t_i$. A lower MSE indicates a better fit to the observed physical reality and thus a more successful discovery.

## C   EXTENDED EXPERIMENTS

### C.1   NOISE ROBUSTNESS STUDY

To evaluate real-world applicability, we injected Gaussian noise into the input trajectories at varying levels of $\sigma$. As shown in Table 6, VIPER-R1 retains high structural understanding even when data is significantly corrupted, whereas baseline VLMs degrade rapidly.

Table 6: Noise Robustness Comparison ($S_{struct}$)

| Noise Level | GPT-5 | Qwen-VL-Max | Gemini-2.5-pro | VIPER-R1-7B |
|---|---|---|---|---|
| 0.05 | 0.31 | 0.34 | 0.30 | 0.78 |
| 0.10 | 0.26 | 0.29 | 0.27 | 0.71 |
| 0.20 | 0.14 | 0.18 | 0.15 | 0.63 |

## C.2    Comparison against Fine-tuned VLMs.

To ensure a fair comparison, we fine-tuned several strong baselines on the PhysSymbol training set. This includes open-source models (Qwen-VL-Max, InternVL3-8B) and the closed-source model (GPT-4o via fine-tuning API).

As shown in Table 7, while fine-tuning improves performance over zero-shot settings, it still lags behind VIPER-R1. This gap highlights that SFT alone is insufficient for mastering the precise topology of physical laws; the reinforcement learning calibration (RGSC) and agentic refinement ($SR^2$) are critical for reaching state-of-the-art performance.

Table 7: Comparison with Fine-tuned VLMs on PhysSymbol. SFT denotes Supervised Fine-Tuning using our MSI curriculum.

| Model | Training Method | $S_{struct}$ | $S_{acc}$ |
|---|---|---|---|
| Qwen-VL-Max | SFT | 0.554 | 0.399 |
| InternVL3-8B | SFT | 0.367 | 0.244 |
| GPT-4o | SFT | 0.621 | 0.488 |
| **VIPER-R1-7B (Ours)** | **MSI + RGSC** | **0.812** | **0.487** |

## C.3    Ablation Studies

To dissect and quantify the contribution of each core component of our framework, we conducted a series of ablation studies on both the 3B and 7B model sizes. We systematically evaluate the performance of: (i) the base Qwen-VL-2.5 model, (ii) the model after only the SFT-based Motion Structure Induction stage, and (iii) our full model, which includes the subsequent RL-based Reward-Guided Symbolic Calibration stage. The results are presented in Table 8.

The results in Table 8 reveal several findings. First, applying MSI alone yields a substantial performance boost over the base model—improving structural scores by over 40 points, which confirms that our two-stage SFT process effectively grounds symbolic reasoning in visual perception and physical intuition. Second, the addition of RGSC further elevates performance across both metrics. For instance, the 7B model's structural score improves from 0.554 (MSI-only) to 0.812 after applying RGSC, and its accuracy score increases from 0.399 to 0.487. Similar trends are observed in the 3B model. These improvements highlight the importance of RL-based symbolic calibration: by optimizing outputs through reward-guided refinement, the model learns to produce more structurally sound and numerically accurate symbolic expressions.

Table 8: Ablation study on the contribution of MSI and RGSC stages for both 3B and 7B models. Each stage provides a significant performance boost.

| Model Size | Model Version | Structural Score ($S_{struct}$) ↑ | Accuracy Score ($S_{acc}$) ↑ |
|---|---|---|---|
| 7B | Qwen-VL-2.5 (Base) | 0.096 | 0.179 |
| | + MSI (SFT only) | 0.554 | 0.399 |
| | **+ MSI + RGSC (Ours)** | **0.812** | **0.487** |
| 3B | Qwen-VL-2.5 (Base) | 0.043 | 0.100 |
| | + MSI (SFT only) | 0.474 | 0.350 |
| | **+ MSI + RGSC (Ours)** | **0.728** | **0.488** |

# D    PhysSymbol: A Comprehensive Multimodal Dataset for Physics Formula Discovery

## D.1    Dataset Overview and Motivation

To train and evaluate our proposed VIPER-R1 framework and provide comprehensive resources for the broader scientific AI community, we constructed **PhysSymbol**, a large-scale synthetic multi-

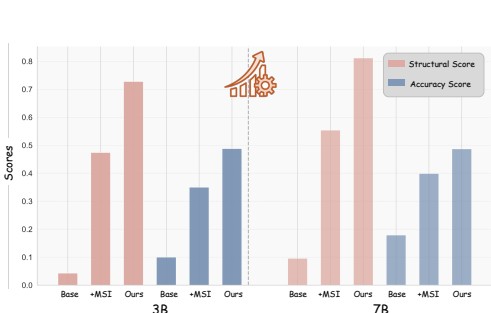 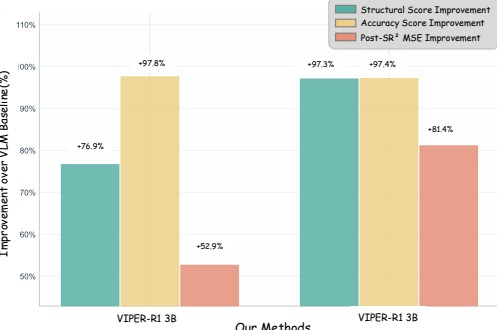

Figure 8: (a) Ablation results show that integrating symbolic regression (+SR²) and our full VIPER-R1 pipeline progressively improves structural and accuracy scores. (b) VIPER-R1 achieves significant relative improvements over zero-shot VLM baselines across all evaluation metrics.

modal dataset that systematically emulates the analytical workflow of physicists studying complex dynamical systems across multiple physics disciplines.

PhysSymbol comprises **10,000 instances** organized into two complementary parts:

- **Part 1 (Training Data):** 5,000 instances focused on classical mechanics and nonlinear dynamics, used for training and evaluating VIPER-R1
- **Part 2 (Research Data):** 5,000 instances spanning five physics disciplines, provided for community research and model evaluation

Each instance contains a complete multimodal representation of a physical system: (1) comprehensive visualizations showcasing differential operator properties, (2) high-resolution numerical field data, (3) ground-truth governing equations, and (4) expert-level causal reasoning annotations for Part 1. This comprehensive design enables frameworks like ours to learn the crucial mapping from visual observations to symbolic mathematical expressions that characterizes human scientific discovery.

### D.2 PART 1: CLASSICAL MECHANICS AND DYNAMICS (TRAINING DATA)

#### D.2.1 PHYSICS TERM LIBRARY AND FORMULA GENERATION

The foundation of PhysSymbol Part 1 lies in a carefully designed physics term library that encompasses the fundamental mechanisms commonly encountered in classical mechanics and nonlinear dynamics. Our term library includes 11 distinct categories of physical phenomena:

**Linear and Nonlinear Restoring Forces:**

- Linear elasticity: $-kx$ with $k \in [0.1, 10]$
- Cubic nonlinearity: $-\beta x^3$ with $\beta \in [0.01, 5]$
- Quintic nonlinearity: $-\delta x^5$ with $\delta \in [0.001, 1]$

**Velocity-Dependent Damping:**

- Linear damping: $-cv$ with $c \in [0.01, 2]$
- Cubic velocity damping: $-\alpha v^3$ with $\alpha \in [0.01, 5]$
- Quintic velocity damping: $-\eta v^5$ with $\eta \in [0.001, 1]$

**External and Coupling Forces:**

- Temporal periodic forcing: $F \sin(\omega t)$ with $F \in [0.1, 5]$, $\omega \in [0.5, 5]$

- Spatial periodic forcing: $F\sin(\omega x)$ with parameters in similar ranges

- Position-velocity coupling: $-\gamma xv$ with $\gamma \in [0.01, 5]$

**Specialized Nonlinear Terms:**

- Trigonometric nonlinearity: $-x\cos(x)$, $-x\sin(x)$ (parameter-free)

- Stochastic perturbations: $\sigma\mathcal{N}(0, 1)$ with $\sigma \in [0.01, 0.5]$

The formula generation process employs a structured combinatorial approach. Each governing equation is constructed by sampling 2-5 terms from the library, with a mandatory linear restoring force to ensure physical stability. Parameters are sampled uniformly from their respective ranges, and the resulting symbolic expression is converted into an executable function for numerical integration.

### D.2.2 HIGH-FIDELITY TRAJECTORY SIMULATION

For each generated governing equation of the form $\ddot{x} = f(x, \dot{x}, t)$, we perform high-resolution numerical integration using the adaptive Runge-Kutta method implemented in `scipy.integrate.solve_ivp`. The simulation protocol follows these specifications:

**Temporal Parameters:**

- Integration duration: $T = 20$ time units

- Sampling resolution: $N = 1000$ uniformly spaced points

- Time step: $\Delta t = 0.02$ (adaptive refinement as needed)

**Initial Conditions:** Initial position $x_0$ and velocity $v_0$ are independently sampled from uniform distributions over $[-1, 1]$ to ensure diversity in trajectory patterns while maintaining numerical stability.

**Data Output:** Each simulation yields a trajectory dataset $\{(t_i, x_i, v_i, a_i)\}_{i=1}^{N}$ containing temporal evolution of position, velocity, and acceleration. This data is exported as CSV files with full numerical precision for downstream analysis.

### D.2.3 DUAL VISUALIZATION STRATEGY FOR PART 1

Part 1 employs a specialized visualization approach tailored for dynamical systems analysis, generating complementary visualizations that capture different aspects of system dynamics:

**Phase-Space Portraits** ($v$ **vs** $x$)**:** These plots encode the kinematic structure and stability properties of the dynamical system. Phase portraits reveal crucial qualitative features such as:

- Closed orbits indicating conservative dynamics

- Spiral trajectories suggesting damped oscillations

- Multiple attractors or limit cycles in nonlinear systems

- Geometric signatures of different restoring force types

**Temporal Trajectories** ($x$ **vs** $t$)**:** These plots emphasize the time-domain behavior and temporal patterns:

- Oscillation frequencies and amplitude modulation

- Exponential growth or decay envelopes

- Periodic forcing signatures and resonance effects

- Transient dynamics and approach to steady states

Both visualizations are rendered as high-resolution images with consistent styling, axis labeling, and grid structures to ensure visual uniformity across the dataset.

### D.2.4 EXPERT-LEVEL REASONING ANNOTATION FOR PART 1

A key challenge in automated scientific discovery lies in bridging the gap between raw visual observation and symbolic reasoning. To address this, the PhysSymbol Part 1 corpus incorporates detailed Causal Chain-of-Thought (C-CoT) annotations, designed to emulate the step-by-step reasoning of a human physicist. These annotations are generated through a structured "Oracle-Annotation" pipeline using GPT-4o. Unlike the inference stage where the model must discover the law from scratch, during this data generation phase, the annotator is provided with the ground-truth symbolic equation and trajectory data to ensure the generated reasoning is factually correct and physically rigorous.

The annotation generation follows a specific three-stage protocol, enforced via system prompts:

- **Stage 1: Phenomenological Observation (Visual).** The prompt first requires the model to describe the qualitative features of the provided plots without jumping to equations. For the Phase-Space Portrait ($v$ vs $x$), it must identify geometric features such as closed orbits (conservation), inward spirals (damping), or limit cycles (non-linearity). For the Temporal Trajectory ($x$ vs $t$), it identifies amplitude modulation, frequency stability, or stochastic irregularities.

- **Stage 2: Physical Mechanism Inference (Physical).** Based solely on the visual observations in Stage 1, the model is asked to infer the underlying physical mechanisms. For example, observing an "exponential decay envelope" in the time domain must be explicitly linked to a "linear damping mechanism"; detecting "frequency dependence on amplitude" must be linked to "nonlinear restoring forces."

- **Stage 3: Symbolic Mapping and Verification (Symbolic).** Finally, the model translates these physical mechanisms into specific mathematical terms (e.g., mapping "linear damping" to the term $-c \cdot v$). Since GPT-4o has access to the ground truth in this generation phase, it performs a self-verification step to ensure the derived logic chain perfectly matches the canonical equation provided in the dataset.

This structured approach ensures that the C-CoT traces are not just generic descriptions, but causal bridges that teach the student model (VIPER-R1) how to derive specific symbolic terms from specific visual cues.

### D.3 PART 2: CROSS-DISCIPLINARY PHYSICS FIELDS (RESEARCH DATA)

### D.3.1 EXPANDED PHYSICS COVERAGE

PhysSymbol Part 2 significantly expands the scope beyond classical mechanics to encompass fundamental physics across five major disciplines, providing a comprehensive testbed for evaluating physics discovery models across diverse domains. This expansion includes:

**Electromagnetic Physics (3 equation types):**

- **Point Charge Field**: $\nabla^2 \phi = -\rho/\varepsilon_0$ (Poisson's equation)
- **Electric Dipole Field**: $\phi = \vec{p} \cdot \vec{r}/(4\pi\varepsilon_0 r^3)$ (Dipole potential)
- **Magnetic Dipole Field**: $\nabla \times \vec{A} = \vec{B}$ (Magnetic field from vector potential)

**Thermodynamics (3 equation types):**

- **Steady Heat Conduction**: $\nabla^2 T = 0$ (Laplace equation)
- **Heat Conduction with Source**: $\nabla^2 T = -S/k$ (Poisson equation)
- **Boundary Layer Temperature**: Thermal boundary layer equation

**Fluid Mechanics (3 equation types):**

- **Potential Flow**: $\nabla^2 \phi = 0, \vec{v} = \nabla \phi$ (Inviscid potential flow)
- **Point Vortex**: $\nabla \times \vec{v} = \Gamma \delta(\vec{r} - \vec{r}_0)$ (Point vorticity)

- **Stagnation Flow**: Linear strain flow equations

**Quantum Mechanics (3 equation types):**

- **2D Harmonic Oscillator**: $(-\hbar^2/2m\nabla^2 + \frac{1}{2}m\omega^2 r^2)\psi = E\psi$ (Schrödinger equation)
- **Free Particle Wavefunction**: $\nabla^2\psi + k^2\psi = 0$ (Free Schrödinger equation)
- **Scattering State**: Cylindrical wave scattering equations

**General Partial Differential Equations (3 equation types):**

- **Diffusion Equation**: $\nabla^2 u - \alpha^2 u = -S$ (Modified Helmholtz equation)
- **Standing Wave Equation**: $\nabla^2 u + k^2 u = 0$ (Wave equation)
- **Helmholtz Equation**: $\nabla^2 u + k^2 u = f$ (Helmholtz equation)

### D.3.2 COMPREHENSIVE DIFFERENTIAL OPERATOR VISUALIZATION

A key contribution of PhysSymbol Part 2 is its systematic generation of eight complementary visualizations for each physics field, providing complete coverage of differential operator properties essential for field analysis:

- **Scalar Field**: Original physical quantity $\phi(x, y)$ distribution
- **Vector Field**: Corresponding vector field $\vec{F}(x, y)$ distribution
- **X Gradient**: $\partial\phi/\partial x$, rate of change in X direction
- **Y Gradient**: $\partial\phi/\partial y$, rate of change in Y direction
- **Gradient Magnitude**: $|\nabla\phi|$, field change intensity
- **Divergence**: $\nabla \cdot \vec{F}$, source/sink distribution
- **Curl**: $(\nabla \times \vec{F})_z$, rotation intensity
- **Laplacian**: $\nabla^2\phi$, field curvature distribution

This visualization approach enables models to learn the relationships between different differential operators and their visual signatures across diverse physics domains, providing a rich foundation for cross-disciplinary physics understanding.

### D.3.3 HIGH-RESOLUTION FIELD GENERATION

Each field in Part 2 is computed on high-resolution $256 \times 256$ grids with domain coverage $[-1, 1] \times [-1, 1]$. Mathematical expressions are implemented with full precision, and parameters are sampled from physically meaningful ranges specific to each equation type. All visualizations maintain consistent color mapping and scaling within each field type to enable systematic comparison and analysis.

### D.4 DATASET ASSEMBLY AND MULTI-FORMAT GENERATION

The final dataset assembly process integrates all components into a unified multimodal format suitable for different training stages and research applications:

**Data Instance Structure for Part 1:** Each complete instance follows the tuple format:

$$(\text{Images: } I_{\text{phase}}, I_{\text{trajectory}}, \text{ Trajectory Data: } M, \text{ Ground-Truth Formula: } E, \text{ C-CoT Reasoning: } C)$$

**Data Instance Structure for Part 2:** Each complete instance follows the format:

$$(\text{Images: } \{I_{\text{scalar}}, I_{\text{vector}}, I_{\nabla x}, I_{\nabla y}, I_{|\nabla|}, I_{\text{div}}, I_{\text{curl}}, I_{\nabla^2}\}, \text{ Field Data: } F, \text{ Equation: } E, \text{ Parameters: } P)$$

**Multi-Stage Training Variants:** To support our three-stage training pipeline, Part 1 generates three dataset variants:

1. **Stage 1 (MSI-Joint):** Full format requiring both reasoning generation and formula prediction

2. **Stage 2 (MSI-Guided):** C-CoT provided as input, only formula prediction required

3. **Stage 3 (RGSC):** Streamlined format for reinforcement learning with structural rewards

### D.5 DATASET STATISTICS AND QUALITY ASSURANCE

The complete **PhysSymbol** corpus consists of 10,000 multimodal instances distributed as follows:

**Part 1 Statistics (Training Data):**

- 5,000 complete instances with trajectory-based visualizations
- Formula complexity: 2-5 terms per equation (average 3.2 terms)
- Coverage: 11 distinct physical mechanism types

**Part 2 Statistics (Research Data):**

- 5,000 complete instances with comprehensive field visualizations
- Coverage: 15 equation types across 5 physics disciplines
- Total files: 85,000 (8 visualizations + 8 data files + 1 info file per instance)
- Purpose: Community research and model evaluation (not used in our training/testing)

To guarantee reliability and usability, we apply comprehensive quality control measures across both parts. Each generated equation undergoes numerical stability verification to avoid degenerate or divergent solutions. Representative samples are visually inspected to confirm clarity and readability of plots. For Part 1, the accompanying C-CoT rationales are validated through automated keyword analysis to ensure coherence with underlying formulas. Finally, file integrity checks are conducted across all multimodal components, ensuring the dataset is complete, consistent, and ready for large-scale experimentation.

The two-part design addresses both the specific needs of training robust physics discovery models (Part 1) and the broader goal of evaluating generalization across diverse physics domains (Part 2). By combining specialized training data with comprehensive evaluation resources, PhysSymbol enables the development and assessment of AI systems that more closely emulate human scientific discovery processes across the full spectrum of physics disciplines.

## E CASE ANALYSIS

In this section, we present detailed qualitative results for several challenging physical systems to provide a more intuitive understanding of the VIPER-R1 framework's capabilities. We first provide an in-depth analysis of a complex non-linear system (Case 1) and then present the visual results for three additional, distinct cases.

### E.1 IN-DEPTH ANALYSIS: NON-LINEAR DAMPING WITH STOCHASTIC NOISE

To showcase our method's full capabilities, we first examine a complex system governed by a linear restoring force, a non-linear damping term, and stochastic noise. The canonical governing equation is of the form $a(t) = -kx - cv^3 + \eta(t)$, where $\eta(t)$ represents a random noise component. This type of system is particularly challenging as it requires identifying and integrating components with fundamentally different mathematical and physical characteristics.

As illustrated in Figure 9, our VIPER-R1 leverages its C-CoT process to perform a sophisticated, physicist-like workflow. It correctly identifies distinct visual cues from the provided plots and maps each one to its underlying physical term.

The model's internal reasoning, extracted from its output, is as follows:

> `<think>` To analyze the given dynamical system, we consider both the $x(t)$ plot and the $v(x)$ phase-space plot. **Key visual features:**

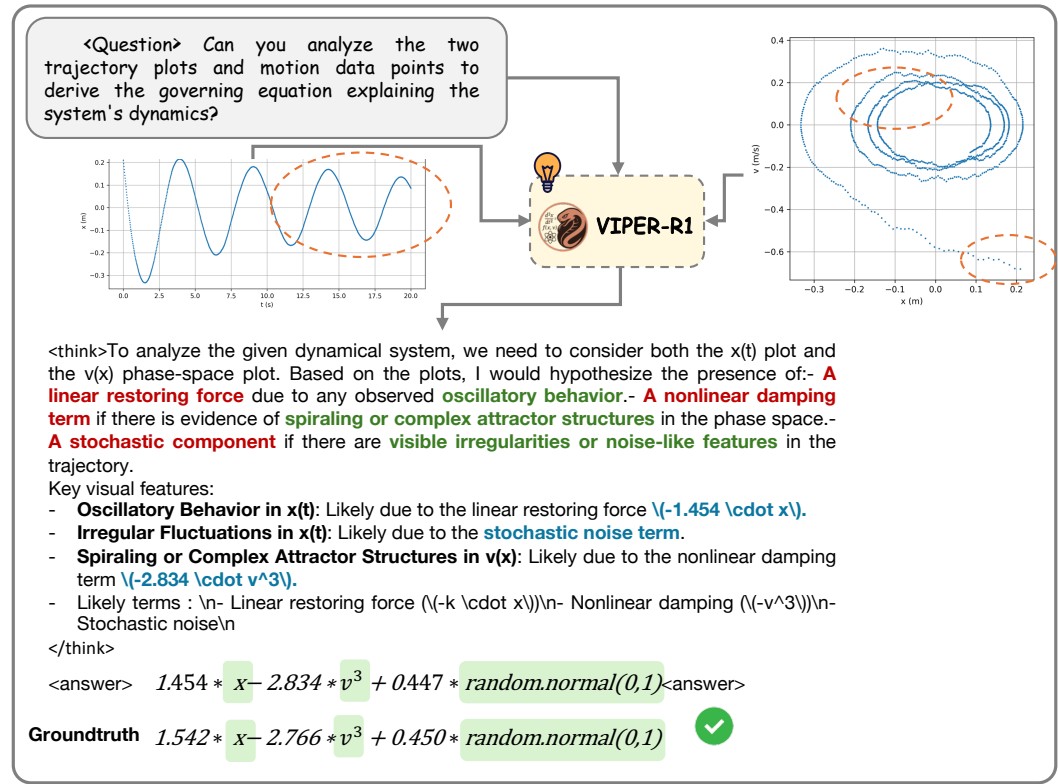

Figure 9: Case 1 Reasoning Process: The VIPER-R1 infers the governing equation of a non-linear dynamical system. Given the x(t) and v(x) plots, the model performs structured visual reasoning to identify key dynamics, including the linear restoring force, non-linear damping, and stochastic noise, before outputting an interpretable symbolic equation.

- **Oscillatory Behavior in x(t):** Suggests a linear restoring force $(-k \cdot x)$.
- **Distorted, Spiraling Attractor in v(x):** Indicates a non-linear damping term, likely dependent on a higher power of velocity $(-c \cdot v^3)$.
- **Irregular, High-Frequency Fluctuations in x(t):** Points to a stochastic noise term.

   **Conclusion:** The system likely combines a linear restoring force, non-linear damping $(-v^3)$, and stochastic noise. `</think>`

This detailed analysis leads the model to propose a hypothesis that is not only structurally correct but also quantitatively close to the true solution, providing an excellent starting point for the subsequent SR² stage. The quantitative success of this process is detailed in Figure 10 and Figure 11, which show the improvements at both the signal and system levels.

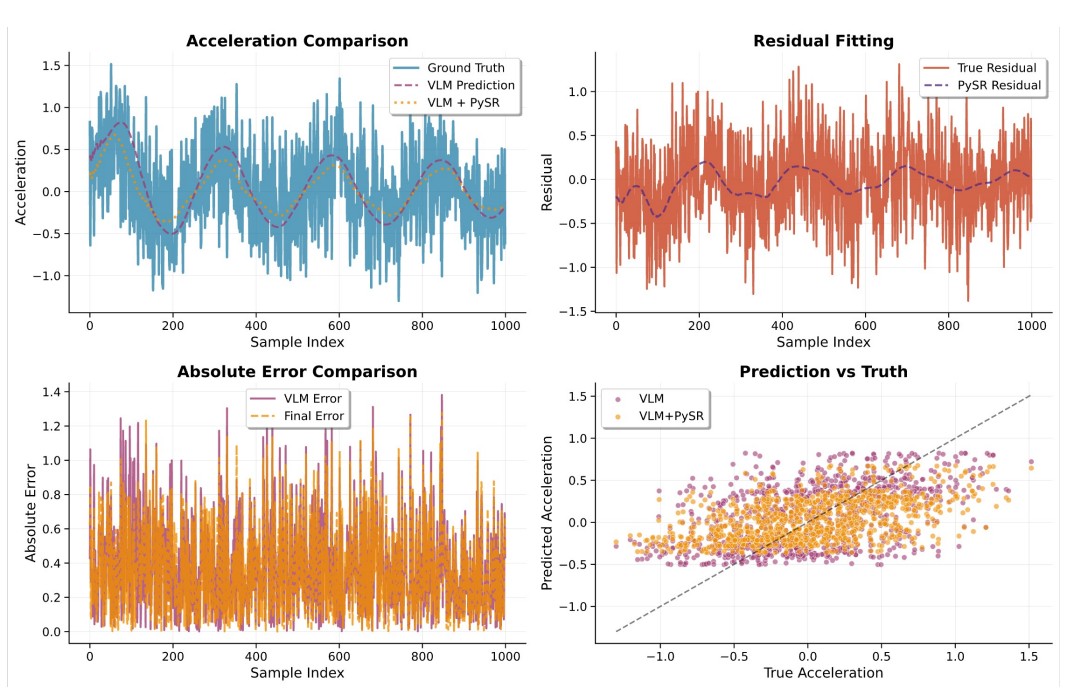

Figure 10: Case 1 Acceleration Signal Evaluation: Comparison of the predicted acceleration signals before (VLM-only) and after (VLM + SR²) symbolic refinement. The refined result demonstrates significantly improved alignment with the ground truth, as shown by the reduced residuals and errors.

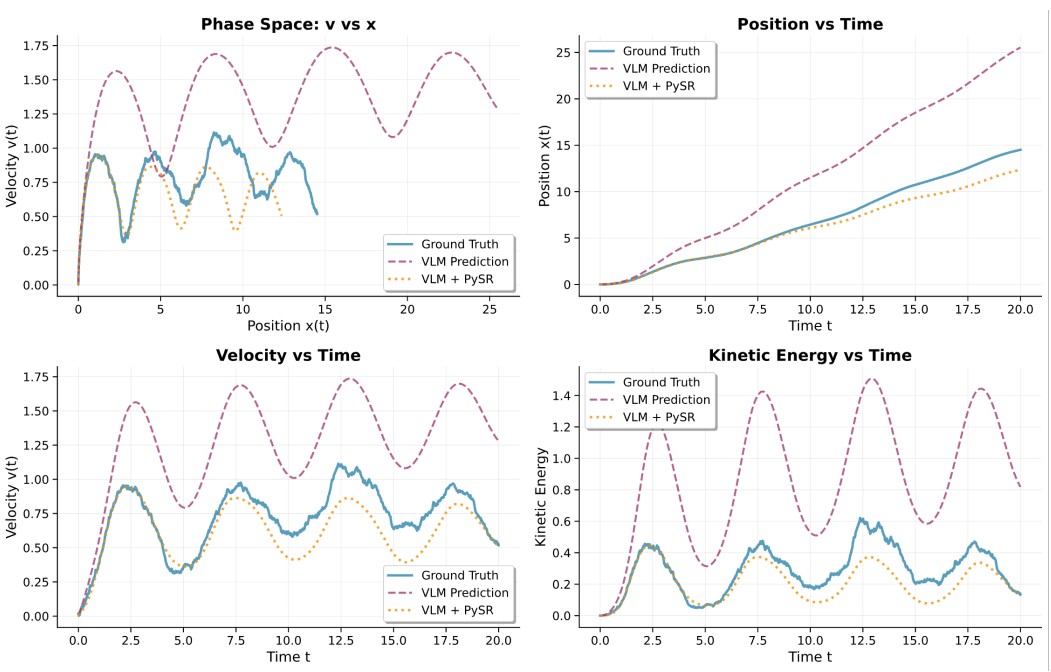

Figure 11: Case 1 System-Level Trajectory Comparison: The phase-space, position, velocity, and energy trajectories generated by the final (VLM + SR²) model show much closer adherence to the ground truth dynamics compared to the raw VLM predictions, indicating physically consistent long-term behavior.

## E.2  ADDITIONAL QUALITATIVE RESULTS

To further demonstrate the robustness of our framework, we present the results for three additional cases, each representing a different class of physical system.

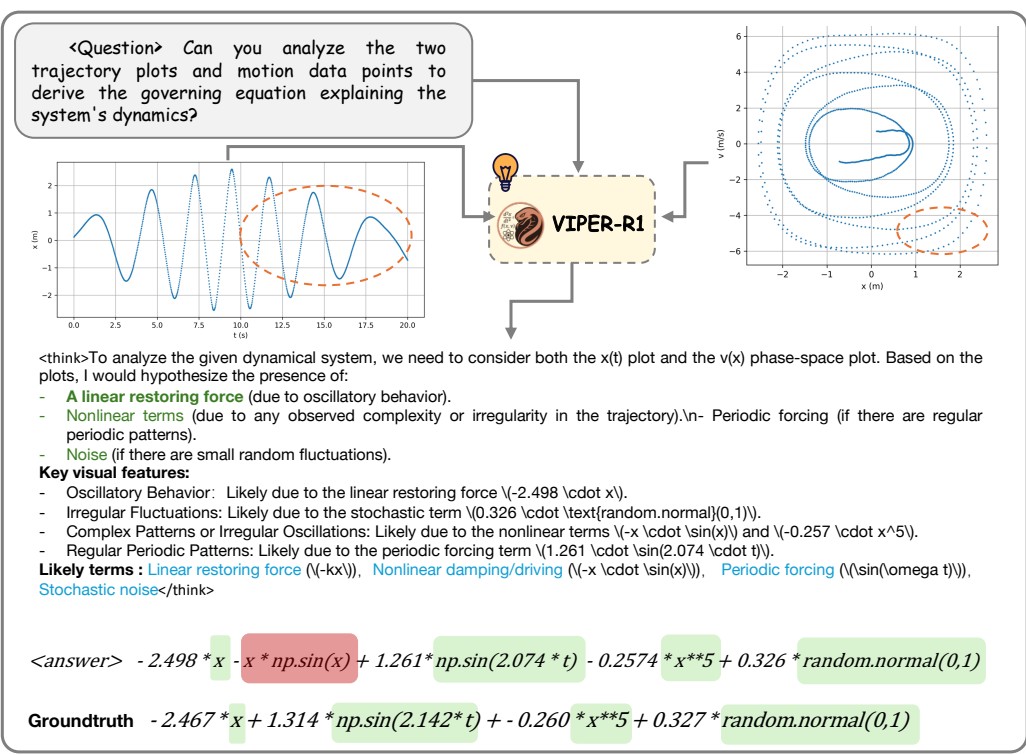

Figure 12: Case 2 Reasoning: A system with linear restoring forces.

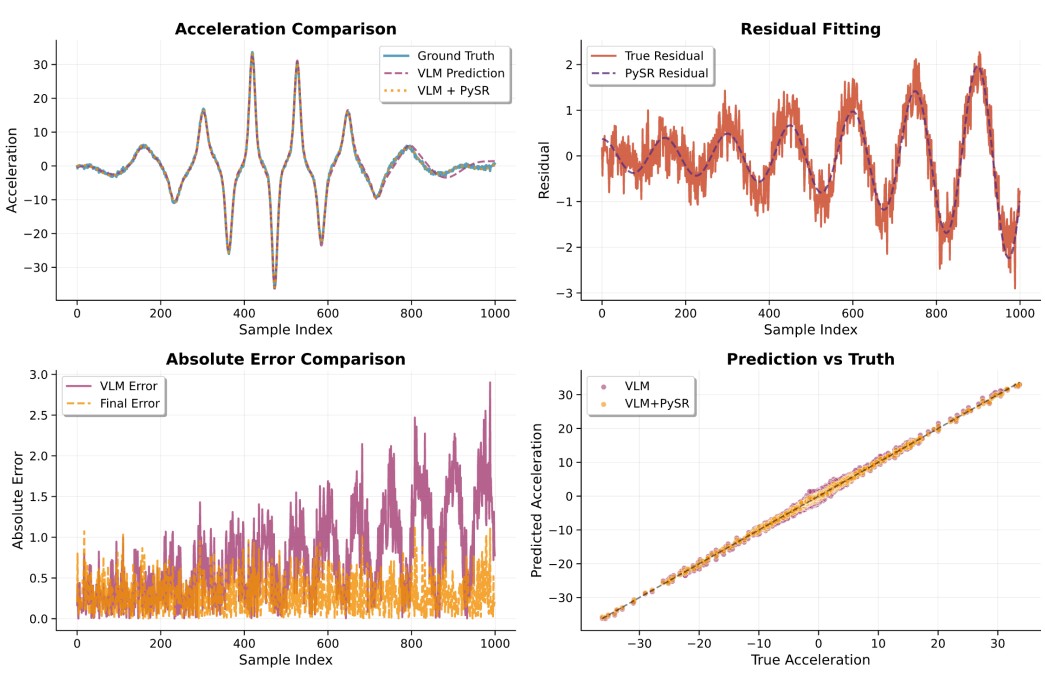

Figure 13: Case 2 Acceleration Signal Evaluation.

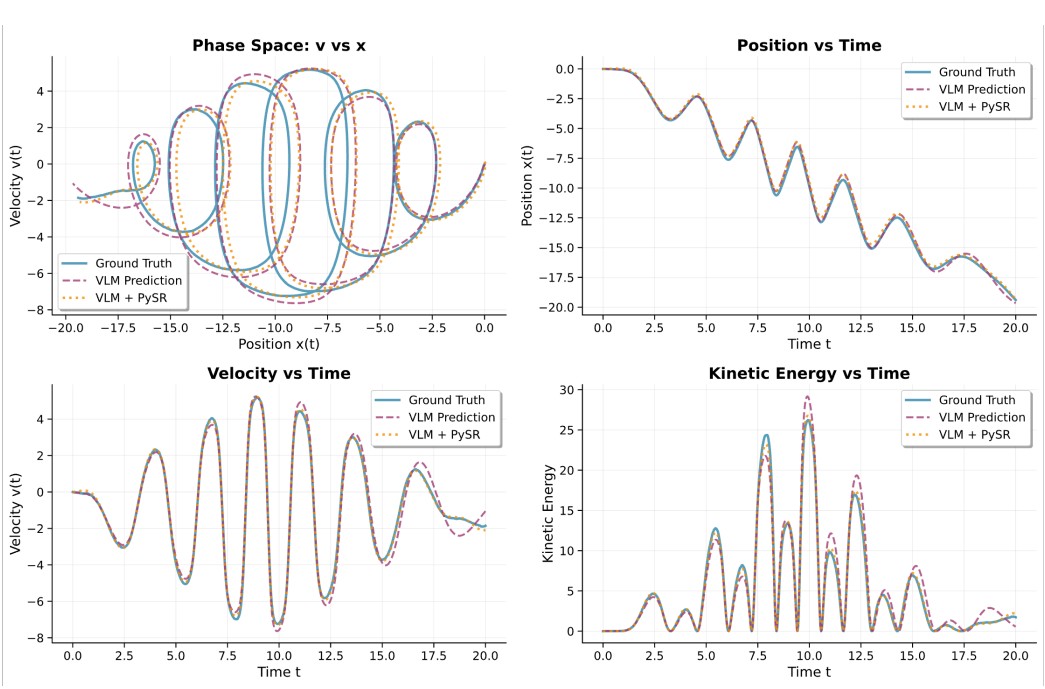

Figure 14: Case 2 System-Level Trajectory Comparison.

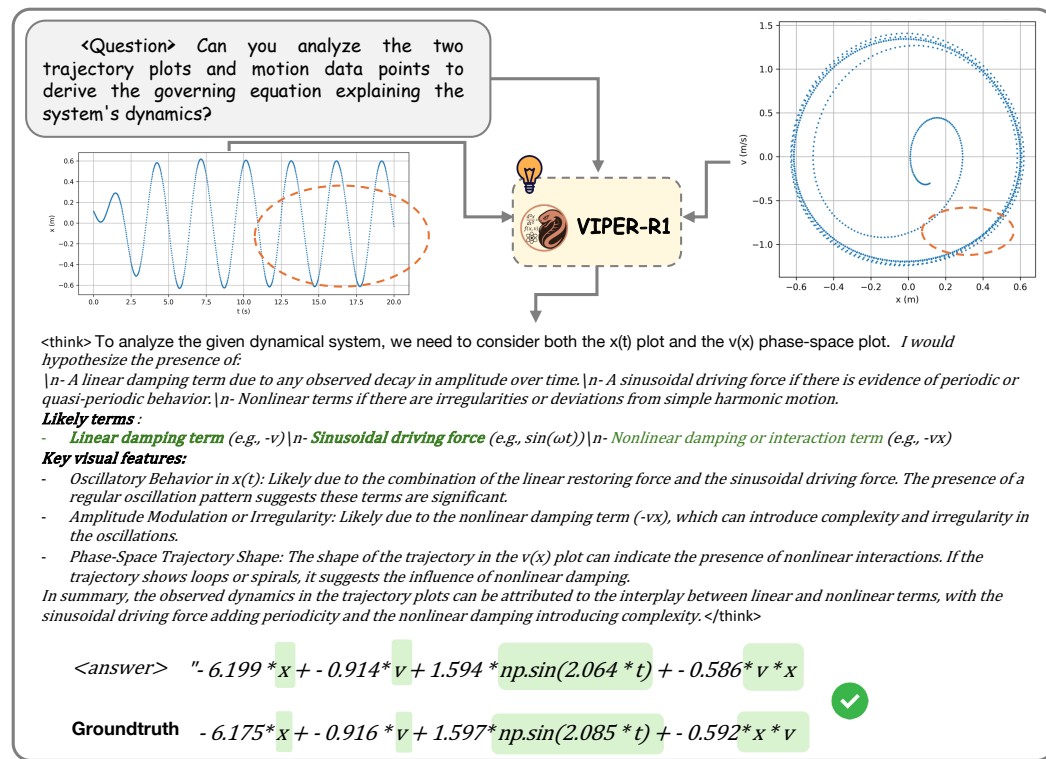

Figure 15: Case 3 Reasoning: A system with sinusoidal driving forces.

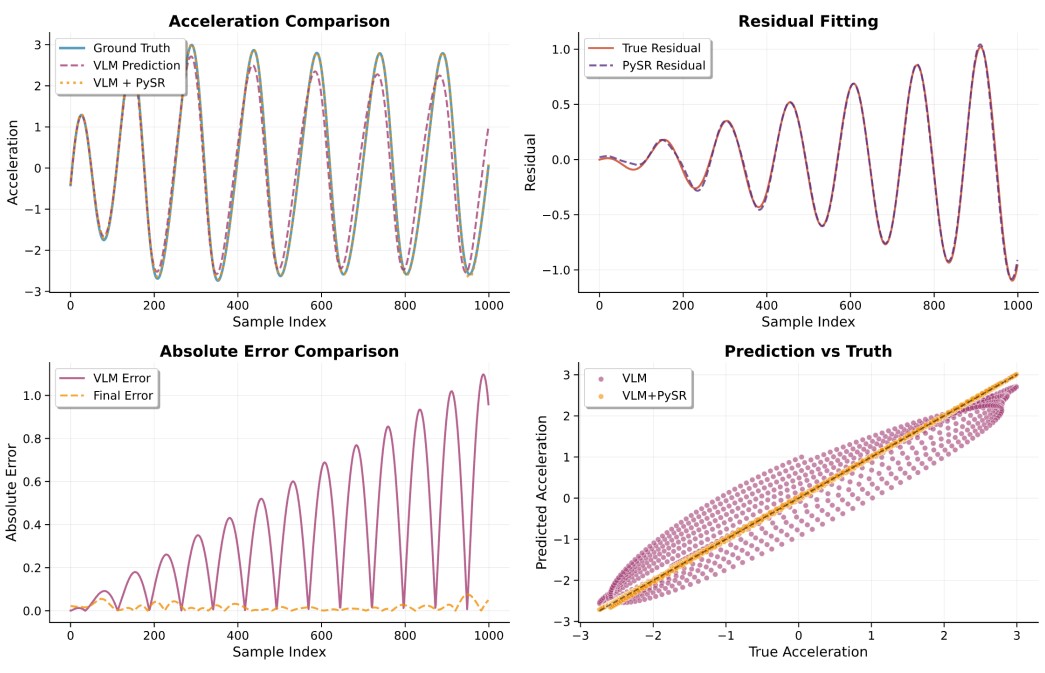

Figure 16: Case 3 Acceleration Signal Evaluation.

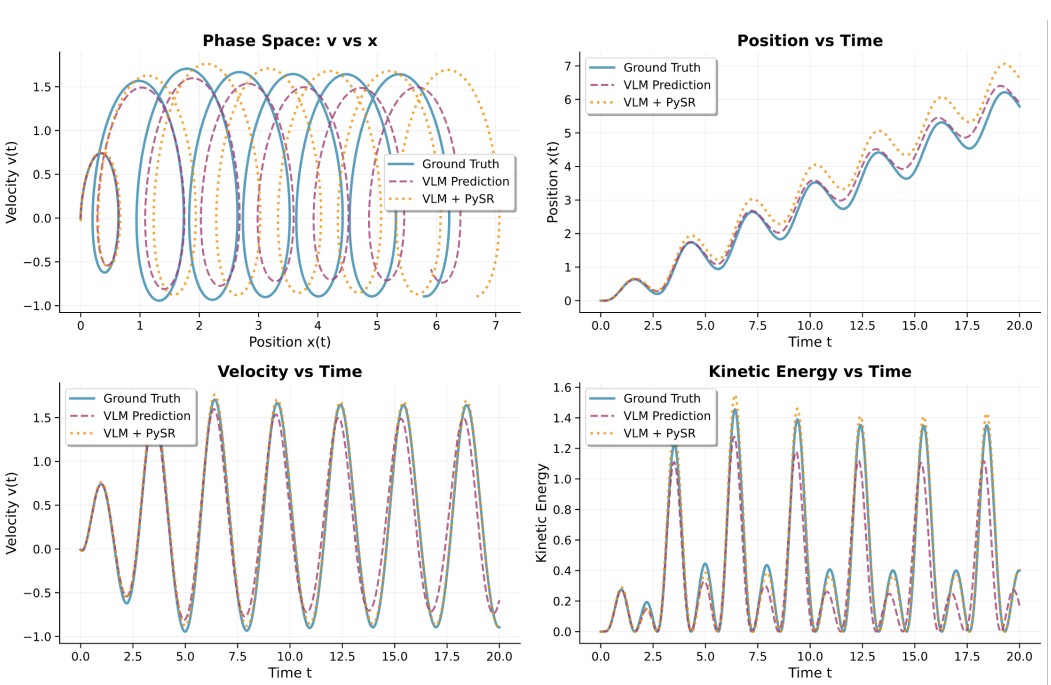

Figure 17: Case 3 System-Level Trajectory Comparison.

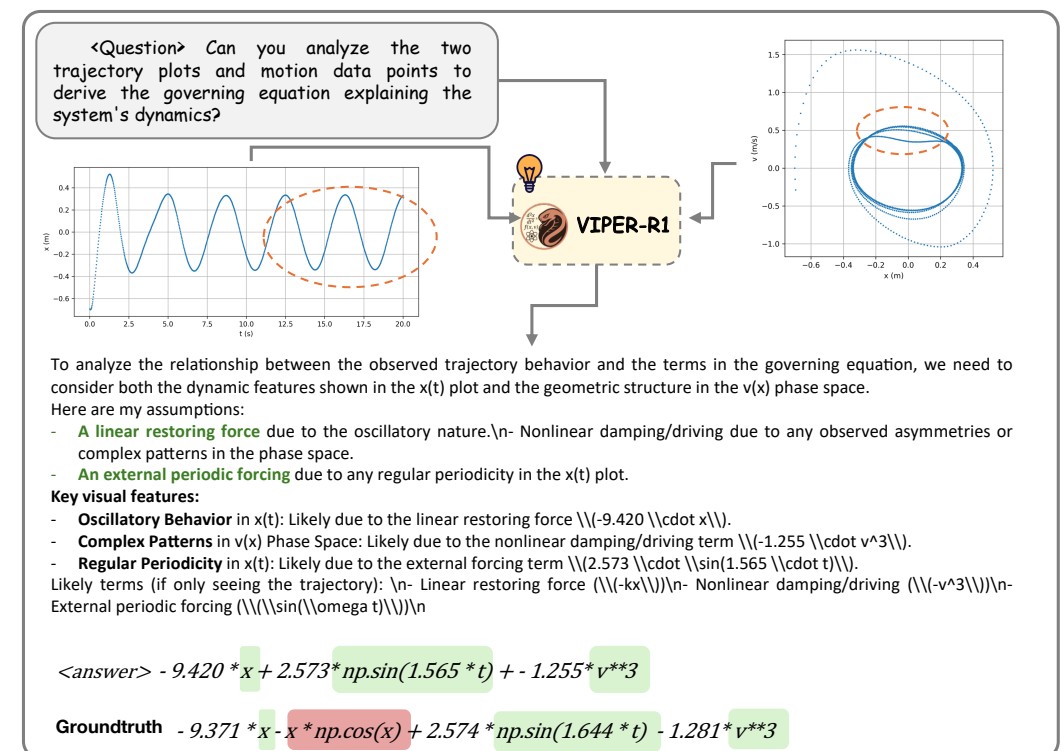

Figure 18: Case 4 Reasoning: A system with external periodic forcing.

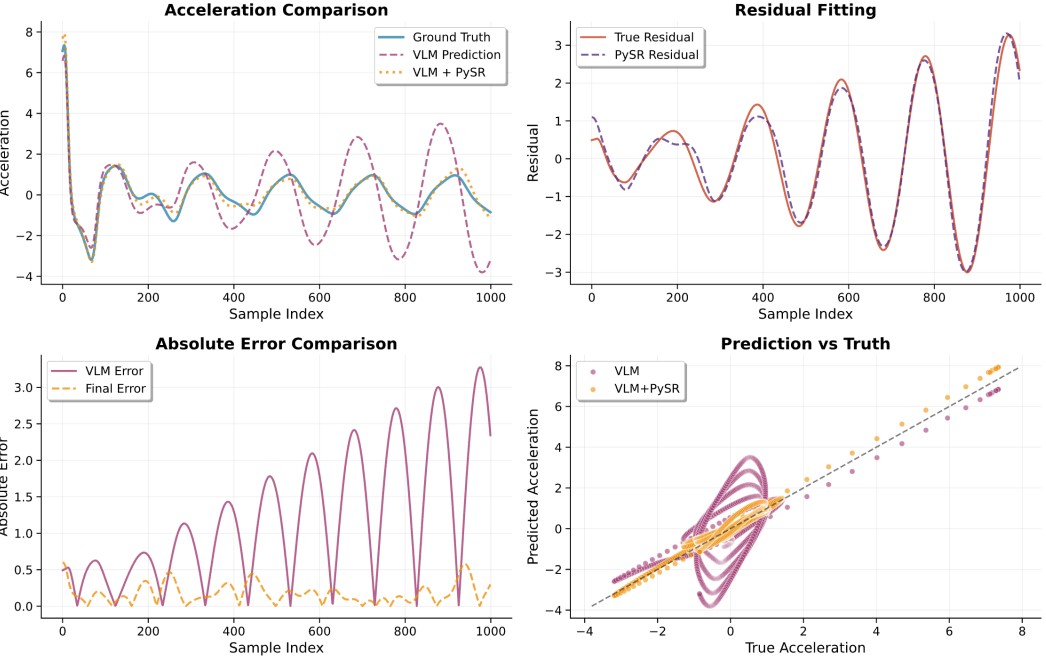

Figure 19: Case 4 Acceleration Signal Evaluation.

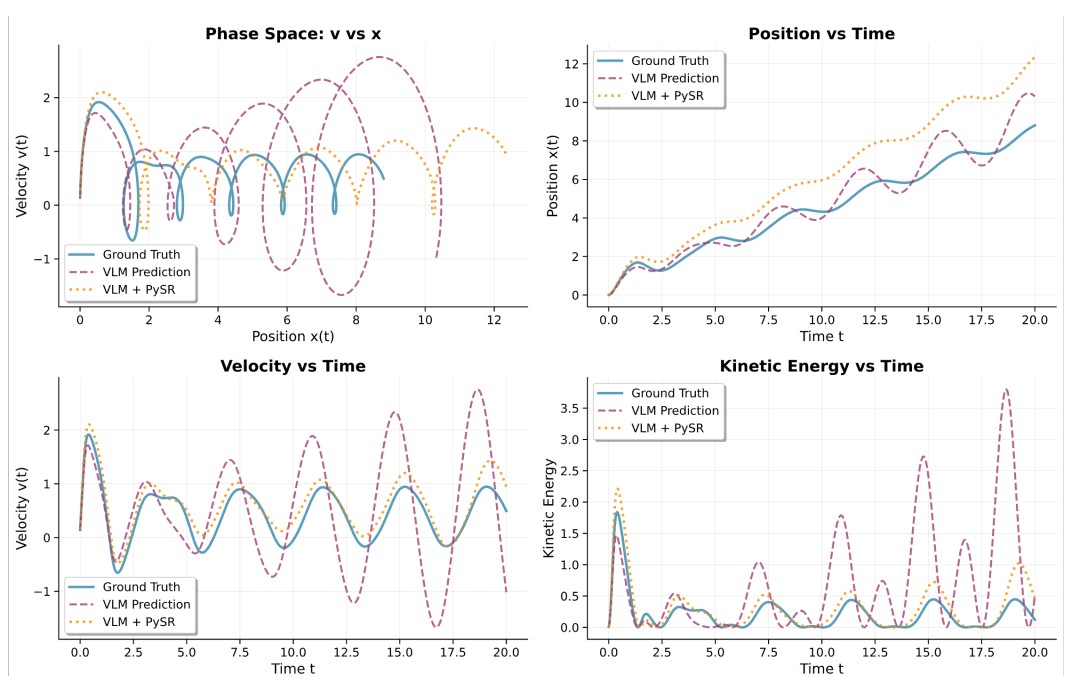

Figure 20: Case 4 System-Level Trajectory Comparison.

## F   THE USE OF LARGE LANGUAGE MODELS

In accordance with ICLR 2026 guidelines, we disclose the use of large language models during the preparation of this paper. We utilized an LLM-based assistant (ChatGPT) exclusively for minor language refinement, including improving grammar, enhancing clarity, and polishing sentence structure in the writing process.

The core research contributions, including the conception of ideas, methodology design, experimental implementation, analysis, and final conclusions, were entirely conducted and authored by the listed human authors. The LLM did not contribute to any part of the scientific content or ideation, nor did it generate original text beyond language-level edits. We take full responsibility for the integrity and accuracy of all contents presented in this paper.

