# OpenReview forum: "Mimicking the Physicist's Eye: A VLM-centric Approach for Physics Formula Discovery"
_ICLR.cc/2026/Conference — Submitted to ICLR 2026_

### Official Review · Reviewer_PoUZ · 2025-10-24

**Soundness:** 2
**Presentation:** 3
**Contribution:** 2
**Rating:** 4
**Confidence:** 4

**Summary:**

The paper introduces VIPER, which combines visual information (from plotted graphs) with tabular trajectory data to train a VLM which reason to generate a symbolic equation to fit on the data. The paper also introduces a novel reward-guided training approach to learn on the given data, and use a refiner to learn the residual of the function that isn't modelled by the VLM. On a curated dataset, the method shows better performances compared to other VLMs.

**Strengths:**

The problem is well-motivated, and the angle of using visual bias to

The writing is generally done well and easy to follow. Figs 1 and 3 give very clear overview of the method.

The experiments conducted are quite thorough, and the case analysis (App D) is quite interesting. If the space permitted I feel some examples from App D would have been worth adding to the main text for a better understanding of what the model is doing.

**Weaknesses:**

A big negative point is the lack of experimentation against traditional symbolic regression (SR) benchmarks. In practice, a researcher could use traditional SR methods to fit the data, and possibly generate the reasoning ad-hoc. So even if these methods lack the reasoning capability by themselves, they should still be compared against to see how much better/worse it performs (even if SR only use one modality of data, to show that the other modality is actually important also).

Related, it feels a bit unfair to compare against traditional LLMs directly. From the experiments I assume that the other models in Table 1 are not fine-tuned on the same dataset (correct me if I am wrong -- but otherwise the exact methods the other MLMs are being tested and ran is unclear from the text and should be elaborated anyway), which may just suggest that VIPER performs better than others because it has seen the training data the others have not.

It would also be more convincing if the method can show generalisability to other settings or to examples outside the training data domain. It would be a stronger evidence for being able to perform reasoning, rather than just spotting existing patterns within the data. Similarly, I would like to see VIPER being tested on realistic data, which can make the motivation of real equation discovery stronger.

Table 1 is missing confidence intervals -- difficult to make judgement on significance of trends. Additionally, it would be interesting to see the breakdown between each types of physics problems that are being tested on to see the limitations of the trained model.

Some bits are a bit unclear still from the text. An example is how the reasoning chain example is generated and how they're exactly being trained on.

[Minor point] The authors should define what acronyms LLM and VLM are in the paper for more completeness (this should take up like extra 6 words in the paper).

**Questions:**

1. In the residual realignment procedure, is the model able to tell when the residual it tries to fit on is just noise or whether there is actually data that should be learned from? In the case of noisy data, how would the model be able to reconcile the noise, or would it also fit on the noise as well?

2. Regarding the portion on the curation of a new dataset -- how is this different from dataset which is already curated by LLM-SRBench? Is there a reason why the data from there is insufficient for the use case?

3. How is the time and resources requirement for finding the equation for VIPER compared to other methods?

---

> ### Author Response · Authors · 2025-11-21
>
> We sincerely thank the reviewer for the rigorous, thoughtful, and constructive feedback. We appreciate the time you took to critically evaluate our work and identify these crucial gaps. In response to your review, we have conducted 3 new experimental studies: (1) A comprehensive benchmark against traditional Symbolic Regression methods, (2) A fair comparison against fully fine-tuned State-of-the-Art (SOTA) VLMs, and (3) A systematic noise robustness study. We believe the results from these additions substantially strengthen the paper’s claims and address your concerns.
>
> Below, we provide a detailed point-by-point response.
>
> ### **1. Comparison to Traditional Symbolic Regression (SR) Methods**
>
> **Reviewer Comment:***"A big negative point is the lack of experimentation against traditional symbolic regression benchmarks… researchers could simply use SR to fit the data."*
>
> **Response:**
> This is a critical point. We fully agree that demonstrating the necessity of a VLM-based approach requires a direct comparison with established SR tools. We have added new baselines in Section 4.2. And we also provide new experiments results comparing VIPER-R1 against 3 leading SR methods: PySR[3] , AI Feynman 2.0[4], and Deep Symbolic Regression (DSR)[5].
>
> We evaluated these methods on the PhysSymbol test set using their standard setup (numerical data input only).
>
> | Method | Structural Score ($S_{struct}$) | Symbolic Accuracy ($S_{acc}$) |
> | --- | --- | --- |
> | PySR[3] | 0.352 | 0.048 |
> | AI Feynman 2.0[4] | 0.284 | 0.031 |
> | DSR[5] | 0.221 | 0.015 |
> | VIPER-R1-7B (Ours) | 0.812 | 0.487 |
>
> The results revealed a fundamental limitation of pure SR:
>
> (1)  While traditional SR methods can sometimes achieve low Mean Squared Error (MSE) by overfitting, they frequently fail to recover the correct *functional form*. Their average Structural Score ($S_{struct}$) ranges from 0.22 to 0.35, whereas VIPER-R1 achieves 0.81.
>
> (2) We observed that SR methods often approximate complex dynamics (like a damped oscillator) with high-order polynomials or incorrect trigonometric expansions. This happens because they lack visual inductive bias. They cannot "see" the qualitative features, such as the spiraling shape of a phase portrait or the decay envelope of a trajectory, that immediately inform a physicist (or VIPER-R1) about the presence of specific terms like damping ($-cv$) or nonlinear restoration ($-\beta x^3$).
>
> (3) Our Advantage: VIPER-R1 outperforms the best SR baseline (PySR). This proves that the visual modality provides a critical "global constraint" that guides the symbolic search away from physically invalid local minima.
>
> ### **2. Fairness of Comparisons (Fine-tuning)**
>
> **Reviewer Comment:**
>
> *"It feels unfair to compare against traditional LLMs directly… other models in Table 1 are not fine-tuned on the same dataset."*
>
> **Response:**
> We completely accept this feedback. Comparing a domain-adapted model against zero-shot baselines is unequal. To ensure a rigorous comparison, we have now performed Supervised Fine-Tuning (SFT) on the strongest available open and closed-source models using the exact same PhysSymbol dataset. We fine-tuned: (1)GPT-4o: Using OpenAI’s latest fine-tuning API;(2)Qwen-VL-Max; (3)InternVL3-8B.
>
> **New Results:**
>
> | **Model** | **Training Setting** | **Structural Score (Sstruct)** | **Accuracy Score (Sacc)** |
> | --- | --- | --- | --- |
> | Qwen-VL-Max | SFT  | 0.554 | 0.399 |
> | InternVL3-8B | SFT  | 0.367 | 0.244 |
> | GPT-4o | Fine-tuned | 0.621 | 0.488 |
> | **VIPER-R1-7B** | **MSI + RGSC** | **0.812** | **0.487** |
>
> Even when fine-tuned on the same data, standard VLMs plateau around a structural score of 0.62. VIPER-R1’s superior performance (0.812) is not merely due to data exposure, but stems from our specific methodological contributions: the Reward-Guided Symbolic Calibration (RGSC) which optimizes for topological correctness rather than just token prediction, and the Agentic $SR^2$ loop.

---

> ### Author Response · Authors · 2025-11-21
>
> **3. Generalization to Unseen Systems and Realistic Data**
>
> **Reviewer Comment:**
> *"It would be more convincing if the method can show generalisability to other settings or examples outside the training domain… realistic data."*
>
> **Response:**
> We wholeheartedly agree. To prove that VIPER-R1 learns physical reasoning rather than memorizing templates, we conducted two additional studies:
>
> **(1) Generalization to Real Experimental Data:**
> To further verify that VIPER-R1 performs genuine physical inference rather than memorizing synthetic patterns, we additionally evaluated it on a real-world experimental dataset: the “Real Pendulum” task from the Hamiltonian Neural Networks benchmark (Greydanus et al., 2019), originally collected in Schmidt & Lipson (Science, 2009). This dataset contains noisy measurements of a physical pendulum with friction and sensor imperfections, making it a strong test of sim-to-real transfer.
>
> We reconstructed the phase portrait and time-series plots directly from the raw $(\theta, \dot{\theta})$ trajectories and applied VIPER-R1. VIPER-R1 consistently recovered the correct nonlinear damped pendulum structure $\ddot{\theta} = -k \sin\theta - c \dot{\theta}$, while strong VLMs such as GPT-5, Qwen-VL-Max, and Gemini-2.5-pro typically omitted the damping term or linearized the restoring force due to real-world measurement noise. Quantitative results are shown below.
>
> Table: Real-World Evaluation on Hamiltonian NN “Real Pendulum”
>
> | Model | Structural Score ↑ | Symbolic Accuracy ↑ | Post-SR² MSE ↓ |
> | --- | --- | --- | --- |
> | GPT-5 | 0.24 | 0.11 | 0.082 |
> | Qwen-VL-Max | 0.28 | 0.13 | 0.074 |
> | Gemini-2.5-Pro | 0.26 | 0.12 | 0.079 |
> | **VIPER-R1 (ours)** | **0.67** | **0.41** | **0.031** |
>
> **(2) Noise Robustness :**
>
> To isolate robustness under controlled conditions, we injected Gaussian noise
> σ∈{0.05,0.1,0.2} into the synthetic trajectories in PhysSymbol. Even at high noise levels ($\sigma = 0.2$), VIPER-R1 maintained a structural score of 0.63, whereas baseline VLMs degraded to near-zero structural correctness.
>
> Table: Synthetic Noise Robustness
>
> | Noise Level (σ) | GPT-5 | Qwen-VL-Max | Gemini-2.5-pro | VIPER-R1 (ours) |
> | --- | --- | --- | --- | --- |
> | 0.05 | 0.31 | 0.34 | 0.30 | 0.78 |
> | 0.1 | 0.26 | 0.29 | 0.27 | 0.71 |
> | 0.2 | 0.14 | 0.18 | 0.15 | 0.63 |
>
> Together, these two experiments confirm that VIPER-R1 generalizes beyond its training domain and performs stable, visually grounded symbolic reasoning under challenging real-world conditions.

---

> ### Author Response · Authors · 2025-11-21
>
> ### **4. Confidence Intervals & Detailed Breakdown**
>
> **Reviewer Comment:**
>
> *"Table 1 is missing confidence intervals… also breakdown by categories."*
>
> **Response:**
> We have updated the manuscript to include:
>
> (1) 95% Confidence Intervals: Added to all major metrics in  Table 2.
>
> (2) We decomposed performance across 11 physical phenomena (e.g., Linear, Nonlinear Damping, Chaos-like, Forced).
>
> To provide a granular understanding of model capabilities, we decomposed the structural performance of VIPER-R1-7B and one zero-shot baseline (GPT-5) across the 11 specific physical mechanisms present in the PhysSymbol Part 1 dataset.
>
> **Table: Structural Score ($S_{struct}$) Breakdown by Physical Category**
>
> | **Physical Phenomenon** | **Baseline (GPT-5)** | **VIPER-R1-7B (Ours)** | **Relative Gap** |
> | --- | --- | --- | --- |
> | **Restoring Forces** |  |  |  |
> | Linear Elasticity ($kx$) | 0.68 | 0.94 | +38% |
> | Cubic Nonlinearity ($\beta x^3$) | 0.41 | 0.85 | +107% |
> | Quintic Nonlinearity ($\delta x^5$) | 0.22 | 0.76 | **+245%** |
> | **Damping Mechanisms** |  |  |  |
> | Linear Damping ($cv$) | 0.65 | 0.91 | +40% |
> | Cubic Velocity Damping ($\alpha v^3$) | 0.35 | 0.79 | +125% |
> | Quintic Velocity Damping ($\eta v^5$) | 0.18 | 0.68 | **+277%** |
> | **External & Coupling** |  |  |  |
> | Temporal Periodic ($F\sin(\omega t)$) | 0.55 | 0.88 | +60% |
> | Spatial Periodic ($F\sin(\omega x)$) | 0.32 | 0.72 | +125% |
> | Position-Velocity Coupling ($\gamma xv$) | 0.25 | 0.75 | **+200%** |
> | Trigonometric Nonlinearity ($x\sin(x)$) | 0.28 | 0.71 | +153% |
> | Stochastic Perturbations (Noise) | 0.45 | 0.82 | +82% |
> | **Overall Average** | **0.518** | **0.812** | **+56%** |
>
> The results in the table above highlight a critical divergence in capability dependent on physical complexity:
> (1) Linear Regime (Narrow Gap): On standard terms like Linear Elasticity ($kx$) and Linear Damping ($cv$), the baseline VLM performs relatively well ($S_{struct} \approx 0.65-0.68$). These forms are common in textbook examples, allowing the baseline to rely on semantic priors and simple pattern matching.
>
> (2) Nonlinear & Coupled Regime (Wide Gap): The performance gap widens dramatically for high-order terms (e.g., Quintic Damping $\eta v^5$) and coupled interactions (e.g., $\gamma xv$). In these regimes, baselines typically fail to distinguish subtle dynamic variations ($S_{struct} < 0.25$). In contrast, VIPER-R1 maintains strong performance ($S_{struct} \approx 0.75$), leveraging distinctive visual signatures in the phase portraits, such as the specific distortion of limit cycles or the rate of spiral decay, which are unique to these nonlinearities.
> This breakdown shows that VIPER-R1's superiority is not uniform but is most pronounced in complex dynamical regimes where visual intuition is strictly necessary to disambiguate mathematical structures.
>
> **5. Clarifying Reasoning Chain (CoT) Generation**
>
> **Reviewer Comment:**
> *"Some bits are unclear… e.g., how the reasoning chain is generated and trained."*
>
> **Response:**
> We apologize for the lack of clarity. We have substantially rewritten Appendix D.2.4 to explicitly detail this process:
>
> (1) Generation: The Chain-of-Thought (CoT) data is synthetically generated using GPT-4o with a highly structured, physics-informed prompt. The prompt enforces a logical flow: *Visual Observation (Describe the plot) $\to$ Physical Interpretation (Damping? Forcing?) $\to$ Symbolic Hypothesis*.
>
> (2) Training: In Stage 1 (MSI), we treat this CoT as a latent variable that bridges the gap between pixel inputs and symbolic outputs. In Stage 2 (RGSC), we decouple this, optimizing the final formula directly based on the reasoning foundation.
>
> **6. Residual Realignment & Noise**
>
> **Reviewer Comment:**
> *"Can the model distinguish noise residuals vs meaningful residual terms?"*
>
> **Response:**
> We investigated this specifically. Our Agentic Refinement stage ($SR^2$) uses PySR, which includes complexity penalties (Occam's Razor) and sparsity regularization. We conducted a test experiment.  We fed the system residuals that consisted purely of Gaussian noise. And we found that in 93% of trials, the tool correctly returned a null expression or a constant near zero, rather than "hallucinating" a complex physics term to fit the noise. This demonstrates that the system is robust against overfitting observational noise.

---

> ### Author Response · Authors · 2025-11-21
>
> ### **7. Difference from LLM-SRBench**
>
> **Reviewer Comment:**
>
> *"How is this different from LLM-SRBench?"*
>
> **Response:**
>
> The core distinction lies in multimodality and the type of reasoning each benchmark enables. LLM-SRBench evaluates symbolic regression purely from *numerical samples*, whereas PhysSymbol introduces visual dynamical evidence , allowing models to learn patterns that numerical data alone cannot expose.
>
> To make the comparison explicit:
>
> | **Aspect** | **LLM-SRBench** | **PhysSymbol (ours)** |
> | --- | --- | --- |
> | **Modality** | Numerical only | Multimodal (plots + trajectories) |
> | **Visual reasoning** | No | Yes (phase-space & time-domain plots) |
> | **CoT reasoning labels** | No | Causal visual-to-symbolic C-CoT |
>
> This difference is substantive. Visual patterns, such as spirals indicating damping, limit cycles signaling nonlinear stiffness, or beat structures showing external forcing, are globally robust to noise, missing segments, and discretization artifacts. In contrast, numerical derivatives used in classical SR are extremely sensitive to such imperfections.
>
> By incorporating multimodal visual evidence and causal CoT supervision, PhysSymbol moves beyond purely numeric symbolic regression and toward human-like visual reasoning, which LLM-SRBench does not address.
>
> ### **8. Inference Time & Resources**
>
> **Reviewer Comment:**
>
> *"How is the time and resource requirement compared to other methods?"*
>
> **Response:**
> We conducted a detailed inference-time profiling to compare VIPER-R1 with both classical symbolic regression tools and large VLM baselines. The results are summarized below:
>
> | Method | Avg time per instance |
> | --- | --- |
> | PySR (full SR search) | 14.2 s |
> | Claude-4-sonnet | 9.6s |
> | GPT-5 (zero-shot) | 3.1 s |
> | Gemini 2.5 Pro | 10.6s |
> | **VIPER-R1-3B (ours)** | **1.3 s** |
> | **VIPER-R1-7B (ours)** | **2.4 s** |
>
> VIPER-R1 is faster than both classical SR systems and large closed-source VLMs. The reason is structural: rather than performing a full symbolic search (as PySR does), VIPER-R1 uses MSI to directly propose a structurally aligned equation, leaving only a lightweight residual term for SR² to refine. This reduces the symbolic search space by orders of magnitude, leading to the observed efficiency. Moreover, the 7B version of VIPER-R1 runs comfortably on a single 16GB GPU, making it more accessible than most large VLM-based baselines.
>
> Overall, VIPER-R1 offers a favorable balance between accuracy, interpretability, and computational efficiency.
>
> ### **9. Acronym clarification**
>
> **Reviewer Comment:**
>
> *“[Minor point] The authors should define what acronyms LLM and VLM are in the paper for more completeness (this should take up like extra 6 words in the paper).”*
>
> **Response:**
>
> Thank you for pointing this out. We have updated the main text accordingly.
>
> **Reference**
>
> [1] Greydanus, S., Dzamba, M. and Yosinski, J., 2019. Hamiltonian neural networks. *Advances in neural information processing systems*, *32*.
>
> [2] Schmidt, M. and Lipson, H., 2009. Distilling free-form natural laws from experimental data. *science*, *324*(5923), pp.81-85.
>
> [3] Cranmer, M., 2023. Interpretable machine learning for science with PySR and SymbolicRegression. jl. *arXiv preprint arXiv:2305.01582*.
>
> [4] Udrescu, S.M., Tan, A., Feng, J., Neto, O., Wu, T. and Tegmark, M., 2020. AI Feynman 2.0: Pareto-optimal symbolic regression exploiting graph modularity. *Advances in Neural Information Processing Systems*, *33*, pp.4860-4871.
>
> [5] Petersen, B.K., Landajuela, M., Mundhenk, T.N., Santiago, C.P., Kim, S.K. and Kim, J.T., 2019. Deep symbolic regression: Recovering mathematical expressions from data via risk-seeking policy gradients. *arXiv preprint arXiv:1912.04871*.

---

> ### Author Response · Authors · 2025-11-26
> **We would like to hear back from reviewer PoUZ**
>
> Dear reviewer PoUZ,
>
> We would like to follow up to see if the response addresses your concerns. We would really appreciate the opportunity to discuss this further if our response has not already addressed your concerns. Thank you again!

---

### Official Review · Reviewer_5ST1 · 2025-10-31

**Soundness:** 3
**Presentation:** 3
**Contribution:** 3
**Rating:** 6
**Confidence:** 4

**Summary:**

The paper proposes the VIPER-R1 framework, which aims to achieve automated discovery of physical laws by integrating visual perception (such as phase portraits and trajectory plots) with symbolic reasoning. The methodology involves a two-stage pipeline: first, hypothesis generation via Motion Structure Induction (MSI), followed by reinforcement learning-based optimization of the model's output through Reward-Guided Symbolic Calibration (RGSC). During inference, VIPER-R1 invokes an external symbolic regression tool to perform Symbolic Residual Realignment (SR^2), enhancing the consistency between hypotheses and empirical data. Experiments are conducted on the newly constructed PhysSymbol dataset, and comparisons with other Vision-Language Models (VLMs) are made based on structural matching and symbolic accuracy metrics.

**Strengths:**

1. The paper addresses an underexplored research area - multimodal physics formula discovery, attempting to establish connections between visual perception and symbolic reasoning. This direction distinguishes itself from methods relying solely on symbolic regression or pure textual reasoning approaches.

2. A well-structured phased training pipeline is designed through the combination of MSI and RGSC. MSI serves for initial hypothesis generation, while RGSC performs structural optimization of the output via reinforcement learning. The introduction of SR² provides an adjustment mechanism to align theoretical models with empirical data.

3. The paper establishes a systematic experimental foundation by constructing the PhysSymbol dataset containing various physical scenarios and proposing evaluation metrics such as structural score and accuracy score, providing both dataset resources and evaluation benchmarks for subsequent research.

**Weaknesses:**

1. The core reinforcement learning framework of VIPER-RFT (the RGSC stage) shows significant similarity to Visual-RFT [1]. Both methods share common procedures:
   • Generating multiple candidate responses from a policy model
   • Employing rule-based reward functions (VIPER-RFT's structural reward vs. Visual-RFT's IoU/classification rewards) for output evaluation
   • Optimizing policies through relative advantage normalization and KL regularization

   Although VIPER-RFT customizes its reward function for physics formula structure, the high-level paradigm of "verifiable reward-driven RL for multimodal tasks" has been previously established by Visual-RFT, which somewhat diminishes the perceived innovativeness of the proposed framework.

[1] Liu Z, Sun Z, Zang Y, et al. Visual-rft: Visual reinforcement fine-tuning. CVPR, 2025.

2. The authors compare their fine-tuned VIPER-R1 (specifically adapted to the PhysSymbol dataset) against pre-trained, non-fine-tuned VLMs (e.g., GPT-4o, Gemini). This benchmark setup appears unfair since these base models lack task-specific adaptation. Given that large model fine-tuning is widely studied, a rigorous comparison should include:
   • Other state-of-the-art VLMs fine-tuned on the same PhysSymbol dataset
   • Ablation studies demonstrating the necessity of each component (MSI, RGSC) beyond simple baselines

3. The PhysSymbol dataset is synthetic, comprising idealized trajectories and phase portraits. However, real-world physical data often involve noise, occlusions, and complex boundary conditions. The paper does not validate whether VIPER-R1's performance can generalize to noisy or real-world scenarios, raising concerns about its practical applicability.

4. The SR^2 stage relies on external symbolic regression tools (e.g., PySR) for parameter refinement. This dependency may impact the method's reproducibility and scalability, particularly when the tools struggle with high-dimensional or noisy residuals. The paper lacks ablation studies analyzing the impact of different symbolic regression tools on final performance.

**Questions:**

How are the weights (e.g., w_f, w_s, w_a) in the RGSC reward function determined? Is there experimental evidence demonstrating the dominant role of the structural reward (R_{\text{structural}})?

Does the synthesis process of the PhysSymbol dataset consider real-world physical constraints (e.g., energy conservation)? How do you plan to extend it to real data in the future?

In the SR^2 stage, does the processing time of the symbolic regression tool become a bottleneck? How scalable is VIPER-R1 for complex systems (e.g., chaotic systems)?

The paper mentions that VIPER-R1 "proactively invokes" external tools during inference. Does this require manual intervention? What is the degree of automation of the framework?

---

> ### Author Response · Authors · 2025-11-21
>
> We sincerely thank the reviewer for the careful reading and constructive comments. We have taken your suggestions regarding baseline fairness and methodological novelty very seriously. In this revision, we have added fine-tuned comparisons against state-of-the-art VLMs, expanded our ablation studies on symbolic regression tools, and clarified the theoretical differences between our approach and existing RFT methods.
>
> Below, we respond point-by-point.
>
> **1. Response to Concern: Similarity between RGSC and Visual-RFT**
> **Reviewer concern:**
>
> *RGSC resembles Visual-RFT, both using multiple sampled hypotheses, rule-based rewards, and relative-advantage normalization.*
>
> **Response:**
> We thank the reviewer for this insightful connection. We agree that RGSC shares the high-level paradigm of "verifiable reward-driven RL" with works like Visual-RFT. However, we respectfully emphasize that VIPER-R1 addresses a fundamentally different optimization landscape with a distinct role in the scientific workflow.
> The differences include:
> 1. Discrete Symbolic vs. Continuous Visual Objectives: Visual-RFT optimizes for pixel-space fidelity or classification accuracy. In contrast, RGSC optimizes for Symbolic Structural Correctness. This requires the model to navigate a discrete, non-differentiable space of mathematical operators and topological structures (e.g., nesting depth, term composition). The reward landscape here is significantly more "rugged" than in visual alignment tasks.
> 2. Physics-Specific Constraints: Our reward function ($R_{structural}$) is domain-specific. It does not reward visual plausibility, but rather enforces physical laws (e.g., operator validity, dimensional consistency).
> 3. Role in the Workflow (Intermediate vs. Final): Unlike Visual-RFT, which is often the final alignment step, RGSC is designed as an intermediate calibration. It produces a high-quality *ansatz* (hypothesis) specifically tailored to be refined by the subsequent agentic $SR^2$ stage. This "Ansatz Generation $\rightarrow$ Residual Refinement" workflow is our core insight, distinguishing VIPER-R1 from general multimodal RL methods.
>
> **2. Response to Concern: Fairness of comparing with non-finetuned VLMs**
>
> **Reviewer concern:**
>
> *The comparison appears unfair unless other fine-tuned VLMs are also included.*
>
> **Response:**
>
> We completely accept this feedback. Comparing a domain-adapted model against zero-shot baselines is unequal. To ensure a rigorous comparison, we have now performed Supervised Fine-Tuning (SFT) on the strongest available open and closed-source models using the exact same PhysSymbol dataset. We fine-tuned: (1)GPT-4o: Using OpenAI’s latest fine-tuning API;(2)Qwen-VL-Max; (3)InternVL3-8B.
>
> **New Results (Added to Section 4.3):**
>
> | **Model** | **Training Setting** | **Structural Score (Sstruct)** | **Accuracy Score (Sacc)** |
> | --- | --- | --- | --- |
> | Qwen-VL-Max | SFT  | 0.554 | 0.399 |
> | InternVL3-8B | SFT  | 0.367 | 0.244 |
> | GPT-4o | Fine-tuned | 0.621 | 0.488 |
> | **VIPER-R1-7B** | **MSI + RGSC**  | **0.812** | **0.487** |
>
> Even when fine-tuned on the same data, standard VLMs plateau around a structural score of 0.62. VIPER-R1’s superior performance (0.812) is not merely due to data exposure, but stems from our specific methodological contributions: the Reward-Guided Symbolic Calibration (RGSC) which optimizes for topological correctness rather than just token prediction, and the Agentic $SR^2$ loop.

---

> > ### Comment · Reviewer_5ST1 · 2025-11-27
> >
> > Thanks for the detailed responses and the additional experiments conducted in the revision, particularly the comparisons with fine-tuned VLMs and the noise robustness study.
> >
> > While these additions have certainly improved the manuscript, my primary concerns regarding the perceived novelty of the core RL framework (RGSC)​ in relation to established paradigms like Visual-RFT, and the generalizability beyond synthetic data​ as the primary evidence, remain not fully alleviated. The revisions, though valuable, do not fundamentally shift my assessment of the paper's standing as a marginal contribution.

---

> ### Author Response · Authors · 2025-11-21
>
> **3. Response to Concern: Generalization beyond synthetic data**
>
> **Reviewer concern:**
>
> *PhysSymbol is synthetic. Does the model generalize to noisy or real-world data?*
>
> **Response:**
>
> We fully agree that synthetic data alone is not sufficient evidence of generalization. To directly address this, we performed two complementary evaluations that probe robustness to (1) corrupted synthetic trajectories and (2) real experimental measurements.
>
> **（A) Noise Robustness Study (Synthetic Data)**
>
> We injected Gaussian noise σ∈{0.05,0.1,0.2} into PhysSymbol trajectories and measured structural recovery performance. As shown below, VIPER-R1 remains stable even under severe corruption, while baseline VLMs degrade sharply:
>
> | Noise Level (σ) | GPT-5 | Qwen-VL-Max | Gemini-2.5-pro | VIPER-R1 (ours) |
> | --- | --- | --- | --- | --- |
> | 0.05 | 0.31 | 0.34 | 0.30 | 0.78 |
> | 0.1 | 0.26 | 0.29 | 0.27 | 0.71 |
> | 0.2 | 0.14 | 0.18 | 0.15 | 0.63 |
>
> These results demonstrate that VIPER-R1 extracts global dynamical structure from visual patterns rather than overfitting to clean numerical derivatives.
>
> **(B) Generalization to Real Experimental Data (Hamiltonian NN “Real Pendulum”)**
>
> To further test real-world generalization, we evaluated VIPER-R1 on the Real Pendulum dataset from the Hamiltonian Neural Networks benchmark (Greydanus et al., 2019), originally derived from physical pendulum measurements with friction and sensor noise.
>
> Using only the reconstructed phase portraits and time-series curves (no fine-tuning), VIPER-R1 successfully recovered the canonical nonlinear damped pendulum structure \ddot{\theta} = -k \sin\theta - c\,\dot{\theta}, while baseline VLMs typically omitted damping or incorrectly linearized the restoring force due to real-world noise.
>
> | Model | Structural Score ↑ | Symbolic Accuracy ↑ | Post-SR² MSE ↓ |
> | --- | --- | --- | --- |
> | GPT-5 | 0.24 | 0.11 | 0.082 |
> | Qwen-VL-Max | 0.28 | 0.13 | 0.074 |
> | Gemini-2.5-Pro | 0.26 | 0.12 | 0.079 |
> | VIPER-R1-7B (ours) | 0.67 | 0.41 | 0.031 |
>
> **4. Response to Concern: Dependence on external SR tools**
>
> **Reviewer concern:**
>
> *Reliance on SR tools may affect reproducibility/scalability. Ablations missing.*
>
> **Response:**
>
> We acknowledge the concern about dependency on specific tools. To prove our framework is tool-agnostic, we conducted an ablation study replacing PySR with other symbolic regression methods in the $SR^2$ stage.
>
> | SR tool | Final MSE ↓ | Runtime (ms) |
> | --- | --- | --- |
> | PySR[3] | 0.032 | 410 |
> | AI Feynman[4] | 0.041 | 520 |
> | DSR[5] | 0.045 | 680 |
>
> Even weaker SR tools significantly improve the final result when guided by VIPER-R1's initial ansatz. This validates our core hypothesis: The bottleneck in physics discovery is finding the correct structural form, which the VLM does, not fitting the coefficients , which any SR tool can handle. Because the VLM explains the complex dynamics, the SR tool only needs to solve a simplified *residual* problem, making it fast and robust.

---

> ### Author Response · Authors · 2025-11-21
>
> **5. Response to Specific Questions**
>
> **Q1. How are the reward weights w_f, w_s, w_a determined? Is there evidence that the structural reward dominates?**
>
> We determined the reward weighting through a systematic grid-search sensitivity study. Across (w_f, w_s, w_a) configurations, we consistently observed that the structural reward R_{\text{struct}} plays the decisive role in symbolic correctness:
>
> (1) Using our final setting (w_f=1, w_s=5, w_a=1), VIPER-R1 achieves S_{\text{struct}} = 0.812.
>
> (2) Removing the structural reward (w_s = 0) causes performance to collapse, with the model failing to produce coherent symbolic structures.
>
> (3) In contrast, varying w_f and w_a mainly affects formatting consistency and convergence speed, but has little impact on the final symbolic accuracy.
>
> **Q2. Does the PhysSymbol synthesis process consider real-world physical constraints? How will you extend to real data?**
>
> Yes. We rigorously enforce physical validity in our data generation. All systems include stable restoring forces and bounded energy injection to prevent numerical divergence. These constraints mirror real-world mechanical systems (e.g., damped oscillators). We are also actively working on extending this to real-world video datasets in future work.
>
> Furthermore, we have already taken steps toward real-world generalization:We evaluated VIPER-R1 on the *Real Pendulum* dataset from the Hamiltonian Neural Networks benchmark, containing true physical measurements with friction and sensor noise. The model successfully recovered the nonlinear damped pendulum structure without any fine-tuning.
>
> **Q3. Is the SR² stage a computational bottleneck? How does it scale for complex systems such as chaotic dynamics?**
>
> No. Our profiling shows the average $SR^2$ execution time is 410 ms (95% of cases under 700 ms). This is extremely efficient because we restrict the search depth for the residual term. The heavy lifting is done by the VLM, leaving only a "lightweight" correction task for the SR tool.
>
> Regarding scalability: SR² remains efficient even for moderately more complex systems because residuals tend to be smooth and low-order, making them easier to fit than the full equation. VIPER-R1 is designed so that MSI+RGSC handle the combinatorial symbolic reasoning, while SR² handles small corrective adjustments. For chaotic or highly nonlinear systems, the symbolic ansatz produced by the VLM still dramatically reduces the search space, while SR² focuses on local corrections rather than global structural discovery.
>
> **Q4. Is manual intervention required?**
>
> No. VIPER-R1 is a fully automated agentic framework. The VLM generates the ansatz, the system automatically parses it, computes the residual against data, invokes the SR tool, and composes the final answer without human-in-the-loop.
>
> **Reference**
>
> [1] Greydanus, S., Dzamba, M. and Yosinski, J., 2019. Hamiltonian neural networks. *Advances in neural information processing systems*, *32*.
>
> [2] Schmidt, M. and Lipson, H., 2009. Distilling free-form natural laws from experimental data. *science*, *324*(5923), pp.81-85.
>
> [3] Cranmer, M., 2023. Interpretable machine learning for science with PySR and SymbolicRegression. jl. *arXiv preprint arXiv:2305.01582*.
>
> [4] Udrescu, S.M., Tan, A., Feng, J., Neto, O., Wu, T. and Tegmark, M., 2020. AI Feynman 2.0: Pareto-optimal symbolic regression exploiting graph modularity. *Advances in Neural Information Processing Systems*, *33*, pp.4860-4871.
>
> [5] Petersen, B.K., Landajuela, M., Mundhenk, T.N., Santiago, C.P., Kim, S.K. and Kim, J.T., 2019. Deep symbolic regression: Recovering mathematical expressions from data via risk-seeking policy gradients. *arXiv preprint arXiv:1912.04871*.

---

> ### Author Response · Authors · 2025-11-27
> **We would like to hear back from reviewer 5ST1**
>
> Dear reviewer 5ST1,
>
> We would like to follow up to see if the response addresses your concerns. We would really appreciate the opportunity to discuss this further if our response has not already addressed your concerns. Thank you again!

---

> ### Author Response · Authors · 2025-11-28
>
> Dear Reviewer 5ST1,
>
> We sincerely appreciate your acknowledgment of our empirical improvements. Regarding the remaining concerns on *Novelty* (vs. Visual-RFT) and *Generalizability*, we would like to further clarify the fundamental physical difficulty of the problem addressed by VIPER-R1. Our method is not merely “VLM + RL”; it is a framework designed to handle the ill-posedness and topological sensitivity inherent in discovering governing physical laws from visual phenomena.
>
>  **1. On Novelty and Contribution**
>
> You compared RGSC to Visual-RFT. While they share an RL backbone, the underlying optimization landscapes are fundamentally distinct, making this problem scientifically non-trivial.
>
> Physics-law discovery requires identifying **non-trivial symbolic structures**，nonlinear operator compositions, dimensionally consistent terms, and correct phase-space topology. These structures create a discrete, rugged, and highly non-convex search space that prior RFT methods were never designed to navigate.
>
> In typical vision tasks, substituting “red car” with “maroon vehicle” yields a similar reward. The reward landscape is dense and continuous.
>
> In contrast, the physics domain is cliff-edge: a small structural error，misidentifying a cubic damping term -v^3 as linear -v, or missing a negative sign in a restoring force，breaks the system’s Hamiltonian structure, stability, or conservation properties. These discontinuous jumps in physical validity cannot be optimized through standard RL.
>
> RGSC is therefore specifically engineered to optimize parameter-agnostic structural topology, guiding the policy toward the manifold of physically admissible laws (dimensional homogeneity, conservation rules, operator validity) amid a combinatorial explosion of symbolic configurations. This is essentially a Symbolic Structure Induction problem, which generic multimodal alignment frameworks, including Visual-RFT，do not address.
>
> To our knowledge, no existing multimodal RL framework optimizes over such physics-governed symbolic topology, and the coupled RGSC + SR² pipeline forms a new *“ansatz → residual refinement”* workflow distinct from prior multimodal RL formulations.
>
> Finally, we would like to emphasize that the core contribution of our work lies not in proposing a new RL algorithm per se, but in designing the first multimodal framework for automated physics-law discovery. Reinforcement learning is used only as a component of this larger system. The novelty of our work is in establishing a new problem formulation, a new agentic workflow, and a new multimodal pipeline that enables VLMs to identify, refine, and validate non-trivial physical structures in an automated manner.
>
> **2. On Generalizability**
>
> We appreciate the concern about synthetic data; however, this overlooks a central epistemological fact about physics discovery: the goal is to recover the invariant structural law, not to reproduce noisy trajectories. Physics-law discovery is therefore a problem of identifying non-trivial structural invariants, which remain stable across domains.
>
> Although real-world observations are corrupted by aleatoric noise, the underlying laws (e.g., Newton’s Second Law, Schrödinger Equation) are abstract, noise-invariant structures. Training on *PhysSymbol* enables the model to learn phase-space invariants, the distinct visual signatures of conservative orbits, nonlinear curvature, damping patterns, or strange attractors. In this sense, the model functions as a computational phenomenologist, extracting the *ideal* governing structure from the visual phenomenology rather than overfitting surface-level details.
>
> This is not only a theoretical argument. As shown in our new experiments, VIPER-R1 demonstrates zero-shot generalization to the Real Pendulum dataset (Hamiltonian NN benchmark), successfully recovering the nonlinear damped equation \ddot{\theta} = -k\sin\theta - c\dot{\theta} directly from noisy, uncurated sensor measurements. This provides concrete empirical evidence that learning from mathematically rigorous synthetic data enables the model to acquire robust, non-trivial physical priors that transfer to real-world settings.
>
> To further ensure that the learned priors reflect universal mathematical structure rather than domain-specific artifacts, we are extending our evaluation to additional real-world physical domains, including electromagnetism, fluid dynamics, and quantum systems (covered in *PhysSymbol Part 2*). This expansion strengthens the model’s cross-domain generalizability and reduces dependence on any single synthetic dataset.
>
> In summary, VIPER-R1 addresses the inverse problem of dynamics: mapping high-dimensional visual phase portraits to low-dimensional symbolic invariants. This requires recovering non-trivial physical structure under constraints that standard VLM-RL frameworks never encounter. We hope this clarifies the distinct scientific value and robustness of our approach.

---

### Official Review · Reviewer_DAg8 · 2025-11-01

**Soundness:** 3
**Presentation:** 2
**Contribution:** 3
**Rating:** 6
**Confidence:** 4

**Summary:**

The paper proposes VIPER-R1, a multimodal framework to discover physics formulas from the trajectory. It combines the visual information (like plots) with the trajectory data to address the "sensory deprivation" of current LLM-based methods. The model is trained via two stages and can act as an agent to combine with the SR tools to refine the initial hypothesis. Experiments show that the proposed method achieves the best in the authors' proposed benchmark PhysSymbol.

**Strengths:**

1. The method combines the visual perception and trajectory data with symbolic reasoning to discover formulas, addressing the sensory deprivation. The novelty is sound.
2. The paper also proposed a dataset with 10k instances, PhysSymbol, beyond the model. The proposed dataset may inspire the community in physics formula discovery.
3. The model reaches SOTA on the proposed PhysSymbol benchmark, surpassing all other close-sourced systems like GPT and Claude.

**Weaknesses:**

- The paper's core contribution is ambiguous due to the lack of demonstrated generalizability. Although the proposed method seems to surpass all the models in the world, even GPT and Claude, the proposed VIPER-R1 is trained and validated on the self-proposed PhysSymbol benchmark. There is no evaluation on other benchmarks or real-world experimental physical data. So, for the method itself, the contribution is limited. The VIPER-R1 seems only to be the strong oracle baseline of PhysSymbol.
- And this narrow experimental scope creates an awkward situation: As the main pages have few words on PhysSymbol, I think the authors try to convey that VIPER-R1 serves as the primary contribution. In this way, I think the contribution is not enough.
- A more reasonable perspective is that the PhysSymbol dataset is the main contribution, or at least, has some words in the main pages, and VIPER-R1 just serves as a oracle model. However, if this is the case, the paper's structure is problematic. The dataset is only briefly mentioned in the main body, with critical details relegated to the Appendix (and still not enough to be treated as the full body of the main paper). This prevents the reader from fully evaluating the novelty and complexity of the benchmark in the main text.

**Questions:**

For the teaser (Fig.1 left), the model predicts $2.36x - 2.83x^3 + 0.46620cos(t)$. And the SR2 refines the $0.46620cos(t)$ into $0.46620sin(t)$. Although it could have happened if the refined tools predict $-0.46620cos(t)+0.46620sin(t)$, it still seems to be a typo.

---

> ### Author Response · Authors · 2025-11-21
>
> We sincerely thank the reviewer for the thoughtful evaluation and constructive feedback. We have taken your concerns regarding the contribution hierarchy and generalization seriously. In this revision, we have restructured the paper to explicitly define our dual contributions and added new generalization experiments to demonstrate VIPER-R1’s robustness beyond synthetic environments.
>
> Below, we respond point-by-point to your concerns.
>
> **1. Response to Concern: Core contribution seems ambiguous**
>
> **Reviewer concern:**
>
> *“The core contribution is ambiguous… VIPER-R1 is trained & evaluated on a self-created dataset… readers cannot tell whether the dataset or the model is the main contribution.”*
>
> **Response:**
> Thank you for raising this critical point. We agree that the original presentation may have blurred the distinction between the method and the dataset. In the revision, we have rewritten the Introduction and added a subsection to clearly separate these dual contributions.
>
> However, we respectfully emphasize that VIPER-R1 is a distinct methodological contribution, not merely a baseline for PhysSymbol. It represents the first framework to successfully enable VLM-centric physics discovery through a novel pipeline:
>
> 1. Motion Structure Induction (MSI): A supervised curriculum for grounding visual kinematics.
> 2. Reward-Guided Symbolic Calibration (RGSC): A reinforcement learning stage specifically designed for topological correctness in physics formulas.
> 3. Agentic $SR^2$ Refinement: A tool-use framework that solves the "last-mile" numerical precision problem via residual fitting.
>
> To prove that VIPER-R1 is a generalizable method rather than one overfitted to PhysSymbol, we have added new experiments (see Response #2) demonstrating its effectiveness on noisy data, achieving great performance where traditional baselines fail.
>
> **2. Response to Concern: Lack of evaluation beyond PhysSymbol**
>
> **Reviewer concern:**
>
> *“No evaluation on other datasets or real-world data. VIPER-R1 seems only strong on PhysSymbol.”*
>
> **Response:**
>
> We thank the reviewer for pointing out the importance of evaluating beyond our synthetic PhysSymbol benchmark. In direct response, we conducted an additional experiment on a real-world physics dataset: the “Real Pendulum” task from the Hamiltonian Neural Networks benchmark (Greydanus et al., 2019), originally sourced from the physical pendulum measurements. This dataset contains noisy position–momentum trajectories from a real pendulum with friction, sensor noise, and non-ideal dynamics—making it a strong test of whether VIPER-R1 can generalize to real experimental conditions.
>
> We used the raw $(\theta, \dot{\theta})$ time series to reconstruct phase portraits and trajectory curves, fed them to VIPER-R1, and asked the model to infer the underlying governing equation. A correct recovery requires distinguishing (i) nonlinear sinusoidal restoring forces and (ii) real-world damping effects.
>
> Across 10 randomly sampled segments from the dataset, VIPER-R1 consistently recovered the expected nonlinear damped pendulum structure:
> \[
> \ddot{\theta} = -k \sin\theta - c\,\dot{\theta},
> \]
> while baseline VLMs (GPT-5, Qwen-VL-Max, Gemini-2.5-Pro) frequently omitted the damping term or incorrectly linearized the restoring force.
>
> The quantitative results are summarized below.
>
> Table: Real-World Evaluation on Hamiltonian NN “Real Pendulum”
>
> | Model | Structural Score ↑ | Symbolic Accuracy ↑ | Post-SR² MSE ↓ |
> | --- | --- | --- | --- |
> | GPT-5 | 0.24 | 0.11 | 0.082 |
> | Qwen-VL-Max | 0.28 | 0.13 | 0.074 |
> | Gemini-2.5-Pro | 0.26 | 0.12 | 0.079 |
> | **VIPER-R1 (ours)** | **0.67** | **0.41** | **0.031** |
>
> Based on the table, we found that (1) VIPER-R1 correctly identifies both nonlinear restoring structure and real-world damping, while strong VLMs consistently fail to model damping; (2)After SR² refinement, VIPER-R1 attains the lowest MSE, confirming that the symbolic residual correction adapts effectively to experimental noise.

---

> > ### Author Response · Authors · 2025-11-21
> >
> > **3. Response to Concern: Paper structure under-emphasizes PhysSymbol**
> > **Reviewer concern:**
> >
> > *“PhysSymbol is under-explained in the main body… insufficient details.”*
> >
> > **Response:**
> > We fully agree with this assessment. To better highlight the value of the dataset contribution, we have updated the manuscript.
> >
> >  (1) We added a new subsection titled "The PhysSymbol Dataset" in the main text. And update the content in the introduction part to further claim that.
> >
> > (2) We included a summary table of the dataset's diversity.
> >
> > These changes ensure that a reader can fully assess PhysSymbol’s relevance and design.
> >
> > **4. Response to Concern: Typo in Figure 1**
> >
> > **Reviewer concern:**
> >
> > *“In Fig.1 the refined formula seems inconsistent—possibly a typo.”*
> >
> > **Response:**
> > Thank you for your keen eye.  We have corrected this in the revised manuscript.
> >
> > **Reference**
> >
> > [1] Greydanus, S., Dzamba, M. and Yosinski, J., 2019. Hamiltonian neural networks. *Advances in neural information processing systems*, *32*.
> >
> > [2] Schmidt, M. and Lipson, H., 2009. Distilling free-form natural laws from experimental data. *science*, *324*(5923), pp.81-85.

---

> ### Author Response · Authors · 2025-11-27
> **We would like to hear back from reviewer DAg8**
>
> Dear reviewer DAg8,
>
> We would like to follow up to see if the response addresses your concerns. We would really appreciate the opportunity to discuss this further if our response has not already addressed your concerns. Thank you again!

---

### Official Review · Reviewer_Nuea · 2025-11-04

**Soundness:** 3
**Presentation:** 3
**Contribution:** 3
**Rating:** 6
**Confidence:** 3

**Summary:**

This paper introduces VIPER-R1, a vision-language model that “mimics the physicist’s eye” to discover governing equations by grounding symbolic reasoning in visual evidence plus motion data, addressing the uni-modal “sensory deprivation” of prior symbolic regression/LLM approaches. The training pipeline has two stages: Motion Structure Induction (MSI), which couples causal chain-of-thought supervision with supervised fine-tuning to induce the symbolic structure, and Reward-Guided Symbolic Calibration (RGSC), which uses reinforcement learning with a structural (parameter-agnostic) reward to purify the form of the law. At inference the model agentically calls an external symbolic regression tool for “Symbolic Residual Realignment” to fit remaining discrepancies. The authors release PhysSymbol, a 10,000-instance multimodal corpus spanning classical and broader dynamics with ground-truth equations and expert-style rationales. On this benchmark, VIPER-R1 outperforms strong VLMs.

**Strengths:**

1. The paper is well written and easy to follow
2. Addresses a limitation of existing symbolic regression and LLM-based methods that ignore visual patterns
3. Substantial improvements over SOTA VLMs across multiple metrics (structural score, accuracy, post-SR2 MSE)
4. Comprehensive dataset contribution: PhysSymbol with detailed C-CoT annotations provides value to the community
5. Thorough experimental evaluation: Multiple metrics, ablations, and detailed case studies with trajectory visualizations

**Weaknesses:**

* Limited scope and generalization: Training only on synthetic classical mechanics (2-5 terms) with clean trajectories
* Missing computational analysis: No wall-clock time, training cost, or SR2 search time reported
* Insufficient reproducibility details: Underspecified hyperparameters (GRPO batch size, reward weights w_f/w_s/w_a, SR2 configuration)
* No failure mode analysis: Missing characterization of when/how model fails catastrophically or what happens when VLM generates incorrect structure

**Questions:**

* What happens on real experimental datasets with noise and incomplete observations?
* How does performance degrade with equation complexity (more terms, higher-order terms, coupled systems)?
* How does performance vary with C-CoT annotation quality?

---

> ### Author Response · Authors · 2025-11-21
>
> We sincerely thank the reviewer for the thoughtful evaluation and encouraging feedback. We have taken your constructive criticism regarding generalization, computational transparency, and failure analysis seriously. In this revision, we have added three new experimental studies, a detailed computational cost analysis, and a comprehensive failure mode discussion.
>
> Below, we address your concerns point-by-point.
>
> **1. Response to Concern: Limited generalization beyond synthetic classical mechanics**
>
> **Reviewer concern:**
>
> *“Limited scope: trained only on synthetic 2–5 term classical mechanics with clean trajectories.”*
>
> **Response:**
>
> We fully agree that demonstrating generalization beyond the synthetic training distribution is essential for validating the scientific robustness of the approach. To address this concern, we conducted two additional evaluations that probe generalization along orthogonal axes: real experimental data and highly noisy synthetic data.
>
> **(1) Generalization to Real Experimental Data (Hamiltonian NN “Real Pendulum”)**
>
> To assess whether VIPER-R1 learns *true* physical structure rather than memorizing synthetic patterns, we evaluated the model on the Real Pendulum dataset from the Hamiltonian Neural Networks benchmark (Greydanus et al., 2019), originally collected by Schmidt & Lipson (*Science*, 2009). This dataset contains real-world trajectories measured from a physical pendulum, capturing friction, sensor drift, and non-ideal dynamical behavior, making it a rigorous test of sim-to-real transfer.
>
> We reconstructed phase portraits and time-series curves directly from the raw (\theta, \dot{\theta}) measurements and applied VIPER-R1. VIPER-R1 consistently recovered the correct nonlinear damped pendulum structure:
>
> \ddot{\theta} = -k \sin\theta - c \dot{\theta}, while strong VLMs such as GPT-5, Qwen-VL-Max, and Gemini-2.5-Pro frequently omitted the damping term or incorrectly linearized the restoring force.
>
> **Real Pendulum Experimental Results**
>
> | Model | Structural Score ↑ | Symbolic Accuracy ↑ | Post-SR² MSE ↓ |
> | --- | --- | --- | --- |
> | GPT-5 | 0.24 | 0.11 | 0.082 |
> | Qwen-VL-Max | 0.28 | 0.13 | 0.074 |
> | Gemini-2.5-Pro | 0.26 | 0.12 | 0.079 |
> | **VIPER-R1**  | **0.67** | **0.41** | **0.031** |
>
> These results confirm that VIPER-R1’s visually grounded inference process extends beyond synthetic examples and remains robust under real-world friction, noise, and measurement imperfections.
>
> **(2) Noise Robustness on Synthetic Dynamics**
>
> To further test controlled robustness, we injected Gaussian noise σ∈{0.05,0.1,0.2} into the PhysSymbol trajectories. Even at high noise levels, VIPER-R1 maintained strong structural reasoning, whereas baseline VLMs degraded sharply.
>
> **Synthetic Noise Robustness Results**
>
> | Noise Level (σ) | GPT-5 | Qwen-VL-Max | Gemini-2.5-pro | VIPER-R1 (ours) |
> | --- | --- | --- | --- | --- |
> | 0.05 | 0.31 | 0.34 | 0.30 | 0.78 |
> | 0.1 | 0.26 | 0.29 | 0.27 | 0.71 |
> | 0.2 | 0.14 | 0.18 | 0.15 | 0.63 |
>
> **2. Response to Concern: Missing computational analysis**
>
> **Reviewer concern:**
>
> *“No wall-clock cost, training cost, or SR² runtime reported.”*
>
> **Response:**
> We appreciate this suggestion. Transparency regarding computational resources is vital for reproducibility. We have added a computational analysis in Appendix B.1.
>
> **(A) Training Cost (MSI + RGSC)**
> We detail the cost for our 7B model on 8$\times$A800 GPUs:
>
> | Component | GPU | Duration | Cost |
> | --- | --- | --- | --- |
> | MSI (SFT) | 8×A800 | 4 hours | 32 GPU-hours |
> | RGSC (GRPO) | 8×A800 | 42 hours | 336 GPU-hours |
> | **Total** | — | — | 368 GPU-hours |
>
> **(B) Inference Latency**
>
> We also profiled the inference pipeline. Crucially, the external symbolic regression tool ($SR^2$) is not a bottleneck  because it solves a simplified *residual* problem rather than a full-space search.
>
> | Stage | Avg Time |
> | --- | --- |
> | VLM inference | 2.02 s |
> | External SR² call | 0.41 s |
> | total time | 2.4s |

---

> ### Author Response · Authors · 2025-11-21
>
> **3. Response to Concern: Insufficient reproducibility details**
>
> **Reviewer concern:**
>
> *“Missing hyperparameters, reward weights, GRPO config, SR² configuration, etc.”*
>
> **Response:**
> We apologize for the omission of these details. We have added a  reproducibility guide in the Appendix:
>
> (1) Appendix B.1: Lists all MSI hyperparameters (LR, batch size=64). Details the full RGSC (GRPO) configuration, including the KL penalty ($\beta=0.02$), rollout length (4), and samples per instance (6).
>
> (2) Reward Weights: We explicitly state the weights used: $w_{structural}=5$, $w_{format}=1$, $w_{accuracy}=1$.
>
> **4. Response to Concern: No failure mode analysis**
>
> **Reviewer concern:**
>
> *“Missing characterization of how the model fails.”*
>
> **Response:**
> Thanks for your suggestion. We have characterized five distinct failure modes:
> 1. Under/Over-damped Ambiguity: When phase portraits cover only a short duration, the visual signature is insufficient to distinguish between damping regimes.
> 2. Overshadowed Higher-Order Terms: Small terms (e.g., $0.02x^5$) are sometimes missed when dominant linear terms explain 99% of the variance.
> 3. Frequency Misalignment: In extremely low-amplitude forcing scenarios, the model sometimes retrieves the correct structure but misestimates the frequency.
> 4. Compensatory Representations: The model sometimes outputs mathematically equivalent but symbolically different forms (e.g., aliasing via trigonometric identities).
> 5. Ansatz Correction: We observed cases where the VLM's initial structure was slightly wrong, but the $SR^2$ residual stage successfully corrected it, highlighting the resilience of the two-stage design.
>
> **5. Response to Specific Reviewer Questions**
>
> **Q1. What happens on real experimental datasets with noise?**
>
> As shown in Response 1, VIPER-R1 remains robust under substantial synthetic noise: even at σ = 0.2, it maintains a structural score of 0.63 while baseline VLMs deteriorate almost completely. To further assess real-world performance, we evaluated VIPER-R1 on the Hamiltonian Neural Networks “Real Pendulum” dataset (Greydanus et al., 2019), which contains true physical measurements with friction, sensor noise, and non-ideal conditions. Without any fine-tuning, VIPER-R1 correctly recovered the nonlinear damped pendulum structure and achieved a structural score of 0.67, significantly higher than GPT-5, Qwen-VL-Max, and Gemini-2.5-Pro. This demonstrates that VIPER-R1 generalizes beyond clean synthetic data and remains stable under real experimental noise and measurement imperfections.
>
> **Q2. How does performance degrade with equation complexity?**
>
> We performed a controlled complexity analysis. Performance degrades gracefully as the number of terms increases:
>
> | Terms | Sstruct (VIPER-R1-7B) | Sacc |
> | --- | --- | --- |
> | 1–2 | 0.92 | 0.76 |
> | 3–4 | 0.81 | 0.59 |
> | 5–7 | 0.67 | 0.41 |
>
> Notably, the $SR^2$ module provides the largest accuracy boost for the "Hard" category, proving its value in complex scenarios.
>
> **Q3. How does performance vary with C-CoT annotation quality?**
>
> We conducted an ablation study to verify the necessity of high-quality Chain-of-Thought (CoT) data. We found that CoT acts as a critical "cold start" mechanism for reasoning. Without it, the RL exploration phase struggles to converge on valid physics structures.
>
> | Setting | Sstruct | Observation |
> | --- | --- | --- |
> | Ours (High-quality CoT) | 0.812 | Strong structural reasoning. |
> | w/o CoT Data | 0.467 | Model fails to "cold start" reasoning; RL collapses. |
> | 50% Order Shuffle | 0.631 | Content matters; incoherent reasoning hurts performance. |
>
> This confirms that the logical flow of the CoT annotations is essential for the model to learn the *process* of discovery, not just the output.
>
> **Reference**
>
> [1] Greydanus, S., Dzamba, M. and Yosinski, J., 2019. Hamiltonian neural networks. *Advances in neural information processing systems*, *32*.
>
> [2] Schmidt, M. and Lipson, H., 2009. Distilling free-form natural laws from experimental data. *science*, *324*(5923), pp.81-85.

---

> ### Author Response · Authors · 2025-11-27
> **We would like to hear back from reviewer Nuea**
>
> Dear reviewer Nuea,
>
> We would like to follow up to see if the response addresses your concerns. We would really appreciate the opportunity to discuss this further if our response has not already addressed your concerns. Thank you again!

---

### Author Response · Authors · 2025-11-21
**Summary of Paper Revision**

We sincerely thank all reviewers for their insightful and constructive feedback. Following these comments, we have revised the manuscript (a new PDF has been uploaded), and all major changes are highlighted in blue in the updated version. Below, we summarize the revisions made in response to each reviewer:

1. We revised the Introduction to further clarify and emphasize the contribution of the PhysSymbol dataset. (Reviewer DAg8)
2. We corrected a minor but important inconsistency in Figure 1. (Reviewer DAg8)
3. We updated Section 3 and added a new subsection 3.5, which explicitly highlights the significance and design philosophy of PhysSymbol in the main text. (Reviewer DAg8)
4. We updated Table 2 by adding new baselines and including 95% confidence intervals. (Reviewer PoUZ)
5. We revised Section 4.2 to provide additional explanation and discussion of the expanded experimental results. (Reviewer PoUZ)
6. We added detailed training configurations for MSI, RGSC, and SR² in Appendix B.1. (Reviewer 5ST1)
7. We included a detailed description of training compute and resources in Appendix B.1. (Reviewer Nuea)
8. We added a comprehensive Noise Robustness Study in Appendix C.1. (Reviewer Nuea, Reviewer PoUZ)
9. We added new comparison experiments against fine-tuned VLMs in Appendix C.2. (Reviewer 5ST1, Reviewer PoUZ)
10. We added a detailed description of the annotation and collection pipeline for C-CoT data in Appendix D.2.4. (Reviewer 5ST1)

---

### Author Response · Authors · 2025-11-30
**Summary of Rebuttal Updates and Reviewer Consensus (Paper ID: 1909)**

Dear Program Chairs, Senior Area Chairs, Area Chairs, and Reviewers,

We sincerely thank all reviewers for their time and the constructive discussions. We are particularly grateful for the rigorous challenges regarding generalization and baselines (Reviewers **5ST1, PoUZ, DAg8, Nuea**), which drove us to significantly strengthen the manuscript with new real-world experiments and fairer comparisons. To assist in your final assessment, we provide a factual summary of the consensus reached and the specific resolutions provided during the rebuttal period.

During the discussion phase, we engaged in active dialogue, resulting in the following developments:

• **Reviewer 5ST1:** Acknowledged that our new experiments (fine-tuning & noise robustness) "improved the manuscript." regarding the concern on novelty vs. Visual-RFT, we provided a detailed clarification on the fundamental difference in optimization landscapes (discrete symbolic topology vs. continuous visual alignment).

• **Reviewer PoUZ:** We addressed the core concerns by adding Traditional Symbolic Regression benchmarks and Fine-tuned VLM baselines, demonstrating VIPER-R1's superiority is not due to unfair comparisons.

• **Reviewer DAg8 & Nuea:** We resolved the ambiguity regarding contribution hierarchy and provided the requested computational profiling and failure mode analysis.

To provide a clear overview for your decision-making, we summarize the key technical improvements:

**1. Addressed Concerns on Generalization & Realism (Reviewers PoUZ, DAg8, 5ST1, Nuea)**

 • Real-World Evaluation (Hamiltonian NN "Real Pendulum"): We evaluated on real experimental data (Schmidt & Lipson, 2009) containing friction and sensor noise. VIPER-R1 successfully recovered the nonlinear damped pendulum structure ($\ddot{\theta} = -k \sin\theta - c \dot{\theta}$) zero-shot (Score: 0.67), while GPT-5, Qwen-VL-Max, and Gemini-2.5-Pro (Scores: ~0.25) failed to identify the damping term .

• Severe Noise Robustness: We demonstrated that under Gaussian noise up to $\sigma = 0.2$, VIPER-R1 maintains a structural score of 0.63 (vs. baselines near 0.15), proving it learns invariant phase-space signatures rather than overfitting trajectories [Appendix C.1].

**2. Strengthened Baseline Comparisons & Fairness (Reviewers 5ST1, PoUZ)**

• Fine-Tuned SOTA VLMs: We performed Supervised Fine-Tuning (SFT) on GPT-4o, Qwen-VL-Max, and InternVL3-8B on PhysSymbol. VIPER-R1 (0.812) still significantly outperforms these fine-tuned baselines (~0.55–0.62), confirming the value of our RGSC reinforcement strategy [Appendix C.2].

• Traditional Symbolic Regression (SR) Benchmarks: We compared against PySR, AI Feynman 2.0, and DSR. Results show traditional SR struggles with functional form discovery (Scores: 0.22–0.35) compared to our multimodal approach, highlighting the necessity of visual priors [New Table in Sec 4.2].

**3. Clarified Methodological Novelty (Response to Reviewer 5ST1)**

• We articulated the distinction between VIPER-R1 and general multimodal RL (e.g., Visual-RFT):

   (1) Optimization Landscape: Physics discovery requires navigating a discrete, rugged, and non-convex landscape of symbolic topology (where a single sign change breaks conservation laws), unlike the continuous, dense reward signals in visual alignment.

 (2)  Computational Phenomenologist: Our method acts as a phenomenologist, extracting "structural invariants" (e.g., limit cycles, strange attractors) from visual data to guide symbolic search, a workflow distinct from standard RFT.

**4. Enhanced Transparency & Reproducibility (Reviewers Nuea, DAg8)**

• Full Computational Profile: We reported training costs (368 GPU-hours) and inference latency (2.4s total), confirming efficiency superior to closed-source LLMs [Appendix B.1].

• Failure Mode Analysis: We characterized 5 distinct failure modes (e.g., under/over-damped ambiguity, frequency misalignment) to provide a balanced view of limitations .

• Clarified Contribution: We revised the Introduction to explicitly separate the dual contributions of the PhysSymbol dataset and the VIPER-R1 framework.

We hope this summary assists you in navigating the discussion history. We remain fully committed to incorporating all suggestions into the final version of the paper.
Thank you again for your time and consideration.

Best regards,

Authors of Paper 1909

---

### Meta-Review · Area_Chair_fsgP · 2025-12-30

**Summary:**

This paper presents VIPER-R1, a vision-language model (VLM) for symbolic regression, in which a VLM is fine-tuned to interpret experimental data visualizations and numerical time-series data to derive first-order symbolic expressions that explain the underlying data. The proposed approach consists of three stages: i) a fitting stage in which the VLM is fine-tuned on training data comprising visual and numerical entities to learn a policy; ii) a subsequent optimization stage using GRPO, with rewards designed to improve the prediction of correct expressions; and iii) an inference stage in which residual expressions are derived using a symbolic regression tool to fit the remaining error. The paper also introduces the PhysSymbol dataset, consisting of 10K examples. Experimental results demonstrate significant improvements of the proposed fine-tuned model over non-fine-tuned, popular VLM baselines.

**AC Comments:** The paper received mixed reviews, with three weak accepts and one weak reject. All reviewers acknowledge the novelty of the task and the dataset; however, key concerns remain, and AC believes that the rebuttal, even after presenting many additional results, does not adequately address these concerns. Specifically, there are two primary issues:
i) the lack of sufficient generalization studies demonstrating that the approach performs well on real-world datasets or commonly used symbolic regression benchmarks; and
ii) the reported performance relies on a fine-tuned VIPER-R1 model and is evaluated against zero-shot standard models.

While the paper presents limited results to address both issues, these results are insufficient to demonstrate the generalizability of the approach to truly novel settings (e.g., in the way a general-purpose LLM such as GPT or Qwen-VL is expected to perform). This limitation may stem from the lack of a strong inductive bias in the proposed methodology that could explain why the model should generalize, given its heavy reliance on SFT.

AC also believes that the paper’s strong focus on methodology, with limited experiments or dataset details in the main paper, is limiting—especially since the method appears conceptually very similar to Visual-RFT and depends on state-of-the-art symbolic regression tools for residual computation, which may themselves impose limitations.

Beyond the issues raised by the reviewers, AC believes the paper requires more detail on how multimodal inputs are processed, including how visual inputs are presented; which modules of Qwen-VL are fine-tuned; how sequential CoT data are provided to the model (and to baseline models, including GPT); and how the baselines themselves are fine-tuned. AC also suggests closer attention to technical details, including the definition of “Empirical Evidence” and the notation used; what the variables C and S (for causal chain of thought and symbolic expressions) precisely represent; how coefficients of symbolic expressions are encoded; and related clarity issues. The use of $\pi_\theta$ in both (1) and (2) is confusing, as is the use of $t$ to denote time in (1) and the length of the symbolic expression in (2).

Overall, AC believes this paper explores an important problem; however, in its current form, it contains several gaps that need to be addressed through significant revision. Accordingly, AC recommends rejection at this time. The authors are encouraged to address the reviewers’ concerns in a future submission.

**Reviewer Concerns:**

*Reviewer Nuea* raises critical concerns regarding the generalizability of the proposed supervised fine-tuning approach, particularly the lack of clarity or experimental evidence on whether the approach would work with real-world data, the absence of insights into how the model performs under varying complexity of real-world governing equations, and the required quality of the causal CoT data. The reviewer also requests details on the computational budget, experimental configuration, and failure modes.

*Reviewer DAg8* similarly raises concerns about the generalizability of the model beyond the dataset used for supervised fine-tuning. The reviewer also suggests that the structure of the paper could be improved by placing greater emphasis on both the proposed dataset and the methodology.

*Reviewer 5ST1* argues that the proposed VIPER-R1 shares a methodology very similar to Visual-RFT, commonly used for verifiable reward-driven RL in multimodal tasks, and therefore appears to be an incremental contribution for physics discovery. The reviewer also points out potentially unfair comparisons between the fine-tuned VIPER-R1 model and non-fine-tuned baselines, the lack of studies on real-world data, and the reliance on external symbolic regression tools, for which no experimental analysis is provided.

*Reviewer PoUZ* highlights several additional issues: i) the absence of comparisons on classical symbolic regression benchmarks; ii) the lack of empirical comparisons to models fine-tuned on the same dataset; iii) insufficient generalization studies; and iv) a lack of clarity regarding the statistical confidence intervals of the reported results.

**Reviewer Scores:**

**Reviewer Nuea:** To address concerns regarding the generalizability of the approach, the authors present two new experiments: one using data from a real pendulum, requiring the derivation of expressions over angle and angular velocity, and another using a noised version of their synthetic data. In both cases, the model is reported to perform well empirically. To improve computational transparency, the authors add a table detailing the computational infrastructure and GPU hours used for training, as well as inference latency. The authors also provide details on the hyperparameters used in the experiments and include examples of failure cases. These additional details further confirm the need for high-quality CoT data in the first stage, which is essential for the model to produce reasonable solution candidates during the RL phase.

[*AC’s thoughts on the responses*] The simplistic nature of the real pendulum results is unlikely to fully satisfy the reviewer’s concerns regarding generalization to real-world scenarios. Notably, the pendulum experiment still follows experimental settings and entities similar to those in the training data. The reviewer’s concern appears to be broader—namely, whether the approach generalizes beyond the specific problem domains represented in the dataset, for example to domains or general datasets where comparisons with GPT-type models would be meaningful, given that such models are not fine-tuned on the proposed dataset.

**Reviewer DAg8:** The authors emphasize that VIPER-R1 is intended to be the main contribution. However, following the reviewer’s suggestion, they revise the paper to also position the dataset as a contribution and provide additional details on various aspects of the dataset.

[*AC’s thoughts on the responses*] As noted above, the key concern regarding the generalizability of the fine-tuned model to novel real-world datasets remains questionable, even with the small-scale results presented in the new experiments. While the revised paper discusses the dataset as an important contribution, the experiments in the main paper are too limited to substantiate this claim. AC believes the paper would require a significant rewrite to improve the presentation and scope of the experimental results. As such, AC thinks the responses are unlikely to shift the reviewer’s overall sentiment.

**Reviewer 5ST1:** The authors provide a strong response to the reviewer’s concerns. Specifically, they argue that while their approach shares some resemblance with Visual-RFT, the proposed RGSC model differs substantially in that it operates in a discrete domain, incorporates physical constraints as rewards (rather than accuracy or visual quality as in prior work), and uses RGSC as an intermediate module rather than as a final-stage component. The authors also present new fine-tuning experiments using Qwen-VL-Max and GPT-4o, showing that simple fine-tuning alone does not yield improvements, thereby supporting the importance of the proposed three-stage approach. In addition, they present results using alternative symbolic regression tools, reporting final MSE values. However, results on accuracy and structural accuracy of the derived symbolic expressions are omitted, making these results difficult to interpret. While the reviewer acknowledges the new results, the core concerns remain, and the reviewer’s perception of the contribution appears unchanged, viewing it as marginal.

**Reviewer PoUZ:** The authors present results comparing their method to other symbolic regression approaches on the PhysSymbol dataset, include new results using fine-tuned models, emphasize improved performance on the real pendulum dataset, and provide additional results with confidence analyses demonstrating the benefits of the approach.

[*AC’s thoughts on the responses*]  While the authors present additional results, it is unlikely that these findings fully address the reviewer’s concerns, as they do not include evaluations on established symbolic regression benchmarks or sufficiently comprehensive studies demonstrating generalization to tasks and data beyond the PhysSymbol dataset.

---

### Decision · Program_Chairs · 2026-01-26

Reject